# Glutamatergic supramammillary nucleus neurons respond to threatening stressors and promote active coping

Abraham Escobedo[1], Salli-Ann Holloway[1], Megan Votoupal[2], Aaron L Cone[1], Hannah Skelton[1], Alex A Legaria[3,4], Imeh Ndiokho[5], Tasheia Floyd[6], Alexxai V Kravitz[1,3,4], Michael R Bruchas[7,8,9,10], Aaron J Norris[1]*

[1]Department of Anesthesiology, Washington University in St. Louis, St. Louis, United States; [2]Department of Medicine, Northwestern University Feinberg School of Medicine, Chicago, United States; [3]Department of Neuroscience, Washington University in St. Louis, St. Louis, United States; [4]Department of Psychiatry, Washington University in St. Louis, St. Louis, United States; [5]Medical College of Wisconsin, Milwaukee, United States; [6]Department of Obstetrics and Gynecology, Washington University in St. Louis, St. Louis, United States; [7]Center for Neurobiology of Addiction, Pain, and Emotion University of Washington, Seattle, United States; [8]Department of Anesthesiology and Pain Medicine University of Washington, Seattle, United States; [9]Department of Pharmacology University of Washington, Seattle, United States; [10]Department of Bioengineering University of Washington, Seattle, United States

*For correspondence: norrisa@wustl.edu

**Abstract** Threat-response neural circuits are conserved across species and play roles in normal behavior and psychiatric diseases. Maladaptive changes in these neural circuits contribute to stress, mood, and anxiety disorders. Active coping in response to stressors is a psychosocial factor associated with resilience against stress-induced mood and anxiety disorders. The neural circuitry underlying active coping is poorly understood, but the functioning of these circuits could be key for overcoming anxiety and related disorders. The supramammillary nucleus (SuM) has been suggested to be engaged by threat. SuM has many projections and a poorly understood diversity of neural populations. In studies using mice, we identified a unique population of glutamatergic SuM neurons (SuM$^{VGLUT2+}$::POA) based on projection to the preoptic area of the hypothalamus (POA) and found SuM$^{VGLUT2+}$::POA neurons have extensive arborizations. SuM$^{VGLUT2+}$::POA neurons project to brain areas that mediate features of the stress and threat responses including the paraventricular nucleus thalamus (PVT), periaqueductal gray (PAG), and habenula (Hb). Thus, SuM$^{VGLUT2+}$::POA neurons are positioned as a hub, connecting to areas implicated in regulating stress responses. Here we report SuM$^{VGLUT2+}$::POA neurons are recruited by diverse threatening stressors, and recruitment correlated with active coping behaviors. We found that selective photoactivation of the SuM$^{VGLUT2+}$::POA population drove aversion but not anxiety like behaviors. Activation of SuM$^{VGLUT2+}$::POA neurons in the absence of acute stressors evoked active coping like behaviors and drove instrumental behavior. Also, activation of SuM$^{VGLUT2+}$::POA neurons was sufficient to convert passive coping strategies to active behaviors during acute stress. In contrast, we found activation of GABAergic (VGAT+) SuM neurons (SuM$^{VGAT+}$) neurons did not alter drive aversion or active coping, but termination of photostimulation was followed by increased mobility in the forced swim test. These findings establish a new node in stress response circuitry that has projections to many brain areas and evokes flexible active coping behaviors.

## eLife assessment

This **important** manuscript investigates the role of a subpopulation of glutamatergic neurons in the suprammamillary nucleus that projects to the pre-optic hypothalamus area in active coping but not locomotor activity. They provide **solid** evidence from experiments using fibre photometry or photo-stimulation during threatening tasks that these neurons allow animals to produce flexible behaviours in response to stress. This work will be of interest to behavioural and systems neuroscientists.

## Introduction

Threat-response neural circuits are conserved across species and have roles in normal behaviors and psychiatric diseases (*LeDoux and Daw, 2018*; *LeDoux, 2014*; *Mobbs et al., 2015*; *Shin and Liberzon, 2010*). Identifying and responding to threatening stressors is critical for survival, but maladaptive changes in underlying neural circuits can contribute to stress, mood, and anxiety disorders (*Ressler and Mayberg, 2007*; *Kessler et al., 2007*). Active coping in response to stressors is a psychosocial factor associated with resilience against stress induced mood and anxiety disorders (*Southwick et al., 2005*). Available evidence indicates that active (e.g. escape, fighting) and passive (e.g. freezing, immobility) coping responses to stressors are governed by separable neural circuits (*LeDoux and Daw, 2018*; *Penzo et al., 2015*; *Ma et al., 2021*). The functioning of circuits underlying active coping could be key for overcoming anxiety and related disorders (*LeDoux and Gorman, 2001*; *Steimer, 2011*).The neural circuits, cells, and mechanisms underlying active coping strategies remain unclear (*Gross and Canteras, 2012*; *Amorapanth et al., 2000*).

The supramammillary nucleus (SuM) has been suggested to be engaged by threatening stressors and has efferent connections to stress-sensitive brain regions and so may be an important regulator of responses to stressors (*Canteras et al., 1997*; *Pan and McNaughton, 2004*; *Vertes and McKenna, 2000*). Research on SuM has focused on connections to the hippocampus and septum, while SuM projections to other brain areas remain less understood (*Pan and McNaughton, 2004*; *Vertes, 1992*; *Kiss et al., 2002*). SuM contains distinct populations with functionally diverse roles including regulation of hippocampal activity during REM, spatial memory, arousal, and environmental interactions (*Kesner et al., 2021*; *Li et al., 2020*; *Billwiller et al., 2020*; *Ito et al., 2018*; *Pedersen et al., 2017*). In addition to functional diversity, anatomical and molecular-cellular diversity is present in SuM (*Leranth and Kiss, 1996*; *Swanson, 1982*; *Kocsis et al., 2003*). Divergent populations in SuM have been defined by differential to the dentate gyrus and CA2 regions of the hippocampus, and neurochemically based on neurotransmitter expression (*Soussi et al., 2010*). Major projections from the SuM, which have yet to be examined, include the preoptic hypothalamus area (POA). We examined if SuM neurons projected to the POA and if this projection could be used to aid in separation of populations in SuM.

We used retrograde adeno associated viral (AAV) and combinatorial genetic tools to identify and characterize a population of glutamatergic SuM neurons with projections to the POA (SuM$^{VGLUT2+}$::POA). We found that SuM$^{VGLUT2+}$::POA neurons represented an anatomical subset of SuM neurons with extensive arborizations to brain regions including those that mediate stress and threat responses including the paraventricular nucleus (PVT), periaqueductal gray (PAG), and the habenula (Hb). Thus, SuM$^{VGLUT2+}$::POA neurons are positioned as hubs with spokes to many areas regulating responses to threatening stressors. We hypothesized this population could respond to and regulate responses to stressors. We found SuM$^{VGLUT2+}$::POA neurons are recruited by multiple types of acute threatening stressors, encode a negative valance, do not promote anxiety-like behaviors, and evoke active coping behaviors. Further, activation of these SuM$^{VGLUT2+}$::POA neurons was sufficient to convert passive coping strategies to active behaviors. These findings indicate SuM$^{VGLUT2+}$::POA neurons are a central hub linked to multiple stress and threat responsive areas and can drive state transitions between passive and active responses to stress.

# Results

## VGLUT2+ SuM neurons projecting to the POA (SuM$^{VGLUT2+}$::POA) arborize to multiple stress-engaged brain regions

The SuM contains functionally diverse and anatomically distinct populations with efferent projections to many brain regions (*Vertes and McKenna, 2000*; *Leranth and Kiss, 1996*; *Haglund et al., 1984*; *Hayakawa et al., 1993*). The diversity includes glutamatergic, GABAergic, and co-expressing GABAergic/Glutamatergic populations (*Billwiller et al., 2020*; *Hashimotodani et al., 2018*). Using retrograde adeno associated virus (Retro-AAV; Retro-AAV2-DIO-tdTomato) and anterograde (AAV5-DIO-ChR2eYFP) tracing, we identified a population of VGLUT2 +expressing neurons in SuM that project to the POA (SuM$^{VGLUT2+}$::POA) with dense projections in the lateral preoptic area (LPO) within the POA (*Figure 1—figure supplement 1A–F*) in VGLUT2-Cre mice (n=4). Using a viral construct encoding for a nuclear restricted fluorophore that switches from mCherry (red) to eGFP (green) in a Cre-dependent manner (Retro-AAV2-Nuc-flox(mCherry)-eGFP), we found SuM$^{VGLUT2+}$::POA neurons were positive for VGLUT2 +in VGLUT2-Cre mice (n=4) and negative for VGAT expression in VGAT-Cre mice (n=3) (*Figure 1—figure supplement 1G–L*) indicating that SuM$^{\textbf{VGLUT2}}$::POA neurons do not belong to GABAergic/Glutamatergic populations.

To examine arborization of SuM$^{VGLUT2+}$::POA neurons, we utilized a combinatorial genetic approach. Specifically, we used mice (VGLUT2-Flp) that express Flp recombinase in VGLUT2 expressing cells in a combination with Flp-dependent expression (fDIO) of Cre and Cre (DIO)-dependent fluorophore expression. We injected Retro-AAV2-DIO-eYFP into POA and AAV-fDIO-Cre in to SuM (*Figure 1A*). We thus expressed Cre in SuM$^{VGLUT2+}$ neurons and, of those, only neurons projecting to the POA (SuM$^{VGLUT2+}$::POA neurons) were labeled with eYFP. We found SuM$^{VGLUT2+}$::POA neurons arborize widely, projecting to multiple brain regions including: the nucleus accumbens (Acb), Septum, lateral hypothalamus (LH), ventral medial hypothalamus (VMH), paraventricular nucleus (PVT), lateral habenula (lHb), the ventral lateral periaqueductal gray (VLPAG), dorsal raphe (DRD), lateral parabrachial nucleus (LPBN), laterodorsal tegmental nucleus (LDTg), and medial vestibular nucleus (MVe; *Figure 1D–M*). We visualized arborization of SuM$^{VGLUT2+}$::POA neurons in cleared tissue by using optical light sheet imaging. For these experiments, we injected VGLUT2-Flp mice (n=3) as in *Figure 1A* and brains were actively cleared using SHIELD for optical light sheet imaging (*Park et al., 2018*). Compiled three-dimensional images of a brain hemisphere, viewed from the medial to lateral perspective (*Figure 1M*) or viewed from the ventral to dorsal perspective (*Figure 1—figure supplement 2M*), showed projections labeled by eYFP from SuM$^{VGLUT2+}$::POA to POA, areas of hippocampus, septum, Acb, and regions in the pons and midbrain. The results further established projections from SuM$^{VGLUT2+}$::POA neurons to multiple brain areas and illustrate the broad arborization.

We used tracing with Retro-AAV's to further corroborate our findings from anterograde tracing by injecting unilaterally into the Acb, septum, PVT, or PAG in VGLUT2-Cre mice. Cell bodies in SuM were labeled eYFP or tdTomato (*Figure 1—figure supplement 2A–L*). Each area was injected in three or more mice, yielding similar results. We examined the anatomic distribution of SuM::POA neurons in SuM by injecting Retro-AAV2-Cre into the POA and AAV5-Nuc-flox(mCherry)-eGFP into the SuM (*Figure 1—figure supplement 3A–B*) of wildtype (WT) Cre- mice (n=3). In mice injected with this combination of viruses, we observed neurons labeled by mCherry (Cre negative) or eGFP (Cre expressing) interspersed in the SuM. We verified the combinatorial viral selectively labeled SuM$^{VGLUT2+}$::POA neurons with minimal background (*Madisen et al., 2010*). As positive controls, we injected Retro-AVV-Flpo into the POA and AAV-fDIO-Cre into the SuM to label the cells in SuM (*Figure 1—figure supplement 3G and H*). As negative controls, we injected only AVV-fDIO-Cre into the SuM in Ai14 mice, which did not yield tdTomato expression (*Figure 1—figure supplement 3C–D*). To confirm that the combination of Retro-AAV2-fDIO-Cres and AAV5-DIO-ChR2eYFP did not lead to labeling of cells with eYFP in the absence of Flp, we injected Retro-AAV2-fDIO-Cre into the POA and AAV-DIO-hChR2eYFP into SuM of WT mice. In these mice we did not observe any expression of eYFP (*Figure 1—figure supplement 3E–F*). The results demonstrate the specificity of our combinatorial viral strategy. All studies were replicated in a minimum of three mice.

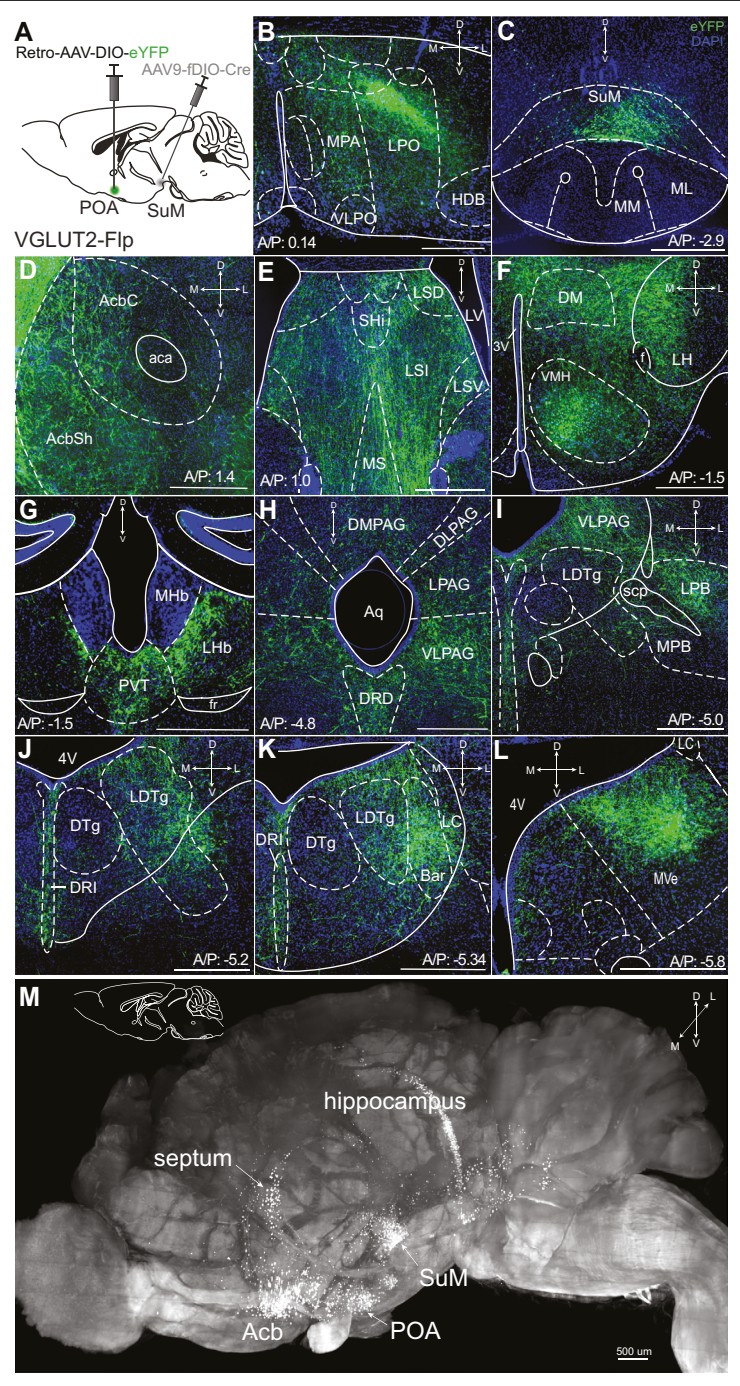

**Figure 1.** SuM^VGLUT2+^::POA arborize widely in the brain. (**A**) Schematic of injections of Retro-AAV-DIO-eYFP into the POA and AAV-fDIO-Cre into the SuM of VGLUT2-Flp mice to label only VGLUT + SuM neurons that project to the POA. (**B**) Projections of VGLUT2 + SuM neurons labeled with eYFP seen in the POA. (**C**) Cell bodies labeled with eYFP in the SuM. (**D–L**) Arborizing processes from the SuM^VGLUT2+^::POA neurons are seen in multiple brain regions including the (**D**) AcbSh and AcbC; (**E**) lateral septum; (**F**) multiple hypothalamic areas; (**G**) lHb and PVT; (**H**) the PAG and the DRD; (**I**) VLPAG and LPB; (**J**) DRI and LDTg; (**K**) and DRI, and MVePC. (**M**) Light sheet microscopy image of a cleared mouse brain hemisphere, viewed from medial to lateral, showing eYFP labeled neurons in SuM^VGLUT2+^::POA with cell bodies in SuM and processes in areas corresponding to septum, hippocampus, Acb, and POA. (500 µm scale bars). Abbreviations- *MPA*- medial preotic, *VLPO*- ventral lateral preoptic, *LPO*- lateral preotic, *HDB*- nucleus of the horizontal limb of the diagonal band *AcbSh*- Accumbens shell, *AcbC*- Accumbens core, *Shi*-septohippocampal nucleus, *LSI*- lateral septal nucleus, intermediate part, *LSV*- lateral septal nucleus, ventral part,

*Figure 1 continued on next page*

*Figure 1 continued*

*LV*- lateral ventricle, *MS*- medial septal nucleus, *lHb*- lateral habenula, *mHb*- medial habenula, *PVT*- paraventricular thalamus, *DM*- dorsomedial hypothalamic nucleus, *LH*- lateral hypothalamic area, *VMH*- ventromedial hypothalamic nucleus, *PAG*- periaqueductal gray, *DMPAG*- dorsomedial periaqueductal gray, *DLPAG*- dorsolateral periaqueductal gray, *LDTg*- laterodorsal tegmental nucleus, *LPAG*- lateral periaqueductal gray, *DRD*- dorsal raphe nucleus, dorsal part, *LPB*- lateral parabrachial nucleus, *MPB*- medial parabrachial, *scp*- superior cerebellar peduncle, *Bar*- Barrington's nucleus, *DTg*- dorsal tegmental nucleus, *DRI*- dorsal raphe, interfascicular part, *LC*- locus coeruleus, *MVe*- medial vestibular nucleus, *4V*- fourth ventricle.

The online version of this article includes the following source data and figure supplement(s) for figure 1:

**Figure supplement 1.** SuM$^{VGLUT2+}$ neurons project to the preoptic area of the hypothalamus and are not VGAT+.

**Figure supplement 1—source data 1.** Quantification of labeling of Cre +and Cre- cells in SuM following injections in POA in VGLUT2-Cre and VGAT-Cre mice.

**Figure supplement 2.** Retrograde verification of projection targets of SuM$^{VGLUT2+}$::POA.

**Figure supplement 3.** Combinatorial viral and genetic approach is effective with minimal background.

## SuM$^{VGLUT2+}$::POA neurons are an anatomical distinct subset of all SuM$^{VGLUT2+}$ neurons

Recent studies have highlighted functionally divergent roles of SuM neurons and suggested differential projections, particularly to regions of hippocampus and PVT, may identify functionally distinct populations (*Kesner et al., 2021*; *Li et al., 2020*). To examine if SuM$^{VGLUT2+}$::POA neurons are a subset of SuM$^{VGLUT2+}$ neurons, we qualitatively compared the projections of total SuM$^{VGLUT2+}$ neurons to SuM$^{VGLUT2+}$::POA neurons. Similar to projections for SuM$^{VGLUT2+}$::POA neurons (*Figure 1*), we observed labeled projections from the total SuM$^{VGLUT2+}$ neurons in the POA, PVT, lHB, and the CA2 field of the hippocampus (*Figure 2A–E*). Importantly, we found areas that received projections from the total SuM$^{VGLUT2+}$ and not from SuM$^{VGLUT2+}$::POA neurons. As schematized (*Figure 2K*), projections from the total SuM$^{VGLUT2+}$ but not SuM$^{VGLUT2+}$::POA populations were present in dentate gyrus (DG) and medial habenula (mHb; *Figure 1G* and *Figure 2F–J*). Connection of the SuM to the dentate gyrus is well described and seen here for the SuM$^{VGLUT2+}$ population (; *Wyss et al., 1979*; *Nakanishi et al., 2001*; *Maglóczky et al., 1994*). The lack of projections from SuM$^{VGLUT2+}$::POA neurons to dentate gyrus, but not to other structures, is a notable difference (*Figure 2J*). Thus, the SuM$^{VGLUT2+}$::POA population represents a subset of the total SuM$^{VGLUT2+}$ neuronal population with distinct projection targets.

## Threatening stressors but not spontaneous higher velocity movement recruits SuM$^{VGLUT2+}$::POA neurons

Neurons in SuM can be activated by acute stressors, and we sought to examine the recruitment of SuM$^{VGLUT2+}$::POA neurons by threatening stressors (*Canteras et al., 1997*; *Day et al., 2004*; *Santín et al., 2003*). We found that forced swimming induced cFos expression in SuM. Labeling SuM$^{VGLUT2+}$::POA neurons using Retro-AAV-DIO-mCherry revealed an increase in the number of mCherry and cFos labeled cells following forced swim (*Figure 3—figure supplement 1A–G*). To further examine recruitment of SuM$^{VGLUT2+}$::POA neurons by acute stressors, we tested a diverse set of acute threatening stressors using in vivo Ca$^{2+}$ detection via fiber photometry. We expressed GCaMP7s in SuM$^{VGLUT2+}$::POA neurons (*Figure 3A–B*) using the combinatorial viral genetic approach detailed for anatomic studies. We developed a dunk assay that utilized a moveable platform allowing mice to be placed into and removed from the water while obtaining fiber photometry recordings. Mice were dunked by lowering the platform below the water level forcing mice to swim for a 30-s trial every 2 min for a total of 10 trials (n=8 mice). During the 30-s swim time, mice exhibited active swimming and climbing behaviors reflected in the quantification of mean time mobile approaching 100% during the swim period without evidence for a shift in the behavioral strategy (*Figure 3—figure supplement 1I*). These data show repeated exposure to an acute stressor, dunk in water, evoked active coping behavior (time mobile). A 95%, 99%, and 99.9% confidence interval (CI) were calculated to analyze the difference in the Ca2+-dependent signal preceding and subsequently after the dunk. Analysis of Ca$^{2+}$-dependent (470 nm excitation) and isosbestic control (415 nm excitation) signals from fiber photometry recordings revealed a significant (99.9% CI) rapid rise of approximately 5 standard deviations in GCaMP Ca$^{2+}$-dependent signal with start of the swim trial that was sustained through the 30-s

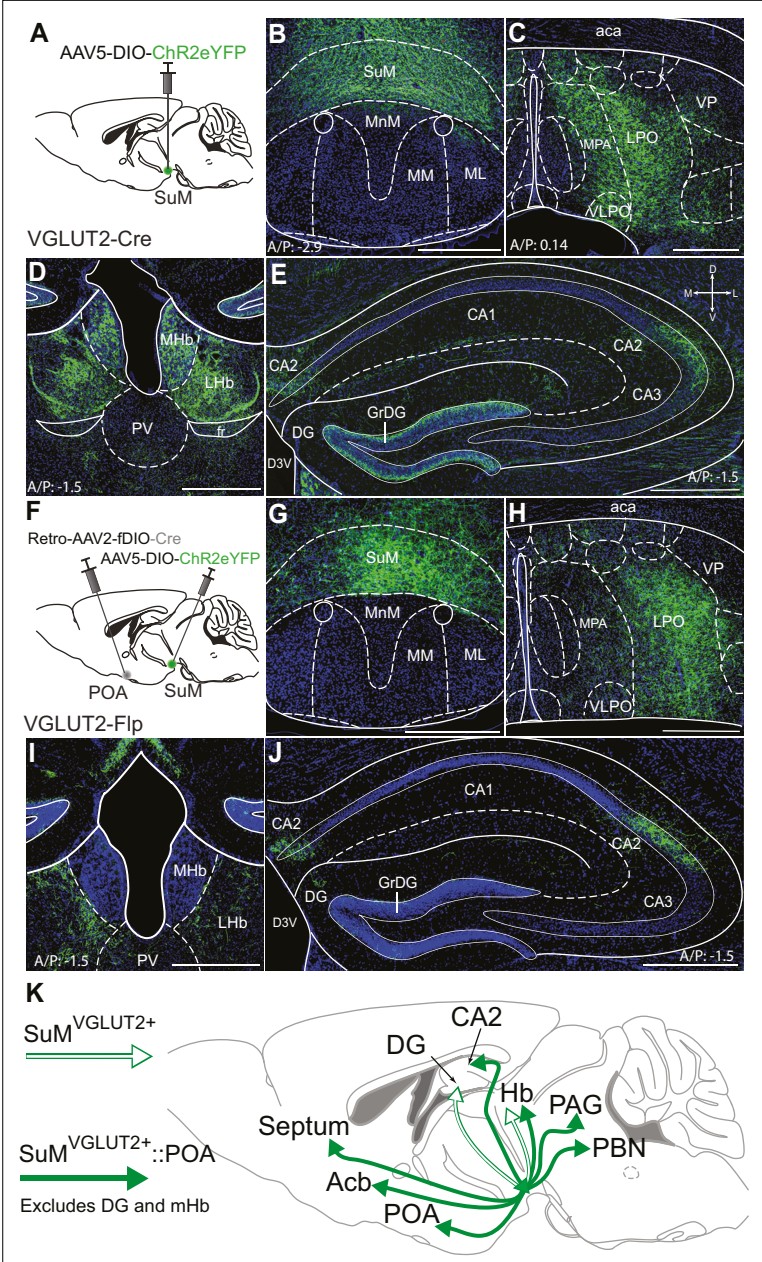

**Figure 2.** SuM^VGLUT2+::POA neurons project to a subset of brain regions compared to all SuM^VGLUT2+ neurons. (**A**) Schematic of injections in VGLUT2-Cre mice of AAV-DIO-ChR2eYFP into SuM. (**B**) eYFP (green) labeled neurons present in SuM and (**C**) processes were observed in the POA. (**D**) Processes from SuM VGLUT2 +neurons were also present in mHb, lHb, and PVT. (**E**) In hippocampus, processes were observed in DG and CA2. (**F**) Schematic of injections in VGLUT2-Flp mice of Retro-AAV-fDIO-Cre and AAV-DIO-ChR2eYFP into SuM. (**G–H**) Cells in SuM and processes in POA were labeled by ChR2eYFP. (**I**) Labeled processes in lHb were evident but none seen in mHb. (**J**) In hippocampus, labeled processes were present in CA2 but not observed in DG. (500 µm scale bars) (**K**) Schematic summary showing the projection to DG and mHb present in the total SuM VGLUT2 +population (outlined arrows) but absent in the SuM^VGLUT2+::POA population (filled arrows).

swim period. The Ca^2+-dependent signal returned to baseline when the platform was raised, removing the animal from the water (*Figure 3F and G*). Repeated exposure to an acute stressor evoking active coping behaviors robustly recruited SuM^VGLUT2+::POA neurons.

To examine additional stressors, we tested a foot shock paradigm (*Figure 3D*) and a predator ambush assay (*Figure 3E*). In foot shock assays, VGLUT2-Flp mice (n=6), prepared as above (*Figure 3A*

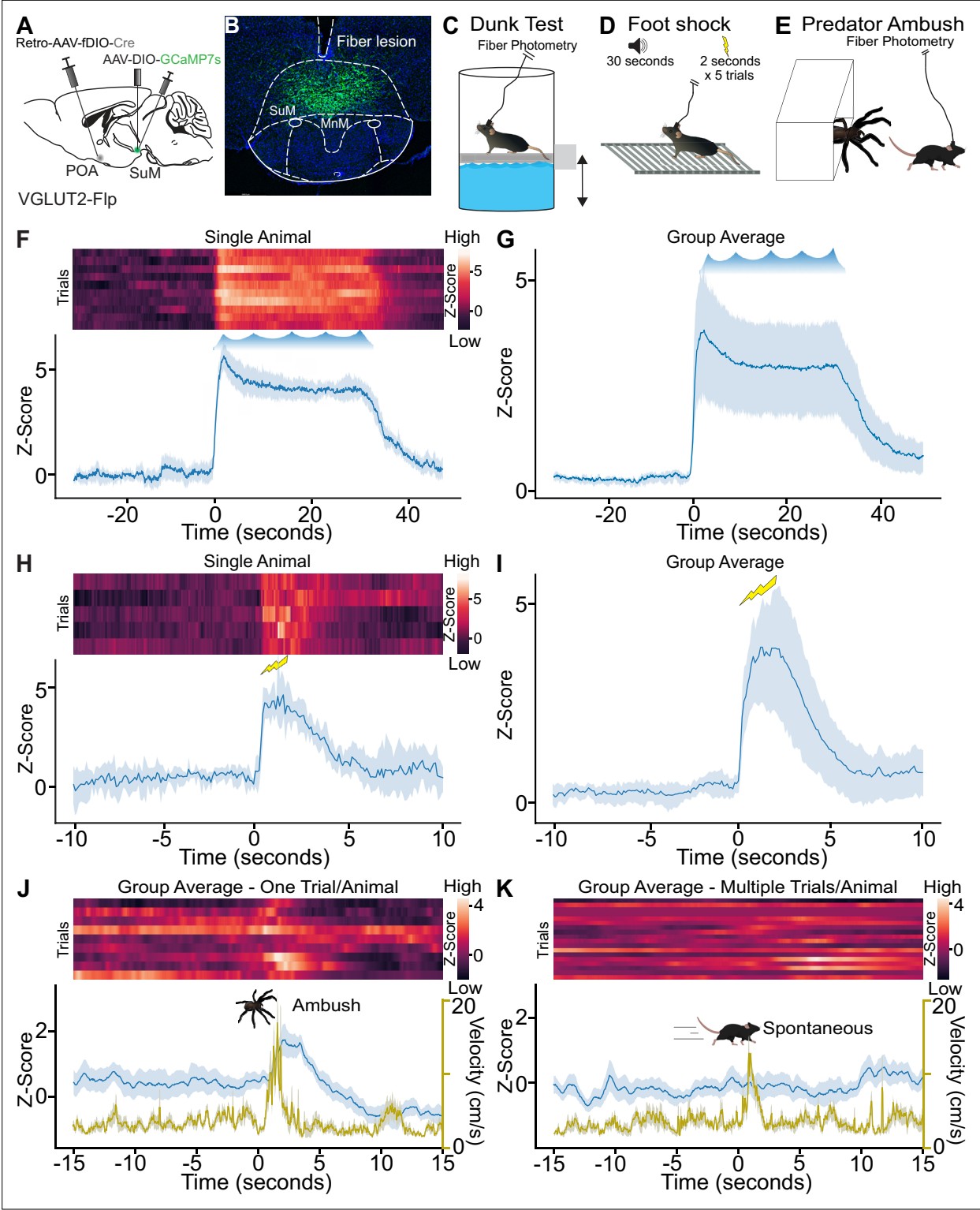

**Figure 3.** Acute stressors recruit SuM^VGLUT2+^::POA neurons. (**A**) Schematic of VGLUT2-Flp mice injected with AAV-DIO-GCaMP7s in SuM and Retro-AAV-fDIO-Cre in the POA with a (**B**) fiber placed over SuM for fiber photometry recordings from SuM^VGLUT2+^::POA neurons. (**C**) While mice were connected for fiber photometry, they were subjected to ten 30 s trials of forced swimming, (**D**) to a 2 s foot shock following a 30 s tone for five trials, or (**E**) to ambush by a mock predator via remote-controlled spider. (**F**) Heat map and mean ± 95% CI Z-score for recordings obtained from a single animal during the repeated forced swim showing increase Ca²⁺ signal during the swim session. (**G**) The mean ± 95% CI Z-score of 10 trials for all animals (n=8) in the dunk assay. (**H**) Heat map and mean ± 95% CI Z-score for recordings from a single animal during the five shock trials showing increase Ca²⁺ signal. (**I**)

*Figure 3 continued on next page*

*Figure 3 continued*

Mean ± 95% CI Z-score for recordings of five trials for all animals (n=6) in the foot shock assay. **(J)** Heat map, mean ± 95% CI Z-score (blue), mean ± 95% CI velocity (gold), for recordings obtained from animals (n=9) during the ambush showing a significant increase (***99.9% CI) in $Ca^{2+}$ signal as the animals flee from the remote-controlled spider with the mean ± 95% confidence interval Z- score of 1 trial (ambush) for all 9 animals. **(K)** Heat map, mean (±95% CI) Z-score (blue), mean (±95% CI) velocity (gold), for the same mice but ambush in the predator assay. Time frame gated for spontaneous locomotion. $Ca^{2+}$ signal does not increase significantly during spontaneous locomotion. Mean peak velocity was not significantly different. For $Ca^{2+}$ signal differences: *=95% CI, **=99% CI, ***=99.9% CI, ns = not significant.

The online version of this article includes the following source data and figure supplement(s) for figure 3:

**Source data 1.** Photometry data for *Figure 3F and G*.

**Source data 2.** Photometry data for *Figure 3H and I*.

**Source data 3.** Photometry data for *Figure 3J and K*.

**Figure supplement 1.** SuM$^{VGLUT2+}$::POA neurons show Fos induction after FST, and dunk test evokes active coping.

**Figure supplement 1—source data 1.** cFos quantification and Dunk FST data.

**Figure supplement 1—source data 2.** This file contains information for cell counts broken out by animal by condition and the underling quantification by animal by trial of time spent mobile in the dunk test.

**Figure supplement 2.** SuM$^{VGLUT2+}$::POA neurons are not recruited during increased locomotor activity in the absence of acute stressor.

**Figure supplement 2—source data 1.** Photometry data and velocity values.

*and B*), were subjected to five pseudorandomly spaced trials with a 30- tone preceding a two-second shock. We observed a significant rapid increase in the $Ca^{2+}$-dependent GCaMP7s signal following the foot shock (99.9% CI; *Figure 3H and I* ). For the predator ambush, we adapted a previously demonstrated paradigm using a mock mechanical spider attack (*Azevedo et al., 2020*). In this assay, a remote-controlled mechanical spider was hidden in a box with a swing door. The box was inside a larger walled arena. Mice were able to freely explore the arena, and at a moment when the mouse was in proximity to the box opening, the spider was moved out toward the mouse. The mice fled, often to a corner, stopped, and turned to face the spider (*Video 1*). In mice (n=9) subjected to the ambush paradigm, we observed a significant (99.9% CI) increase in $Ca^{2+}$-dependent GCaMP signal at the time of the ambush followed by suppression below the initial baseline (*Figure 3J*). Populations of neurons in SuM have previously been found to correlate with future movement velocity (*Farrell et al., 2021*), so we examined if recruitment of SuM$^{VGLUT2+}$::POA neurons was correlated to periods of spontaneous higher velocity movement. Analyzing data collected prior to the ambush event, we examined if increased velocity, at speeds similar to the fleeing induced by the ambush, was correlated with increased $Ca^{2+}$-dependent GCaMP signal in SuM$^{VGLUT2+}$::POA neurons (*Figure 3L*). We found no correlation (*Figure 3K*). We further examined the data for a correlation of increased $Ca^{2+}$-dependent signal in SuM$^{VGLUT2+}$::POA neurons to movement velocity during open field exploration in a sperate cohort of mice (n=13). Here, we found no evidence for correlation of velocity with Z-score of $Ca^{2+}$-dependent signal including a cross correlation analysis to account for a potential temporal offset (*Figure 3—figure supplement 2*). In aggregate, the data support recruitment of SuM$^{VGLUT2+}$::POA neurons by diverse threatening stressors but not during times of higher velocity spontaneous movement.

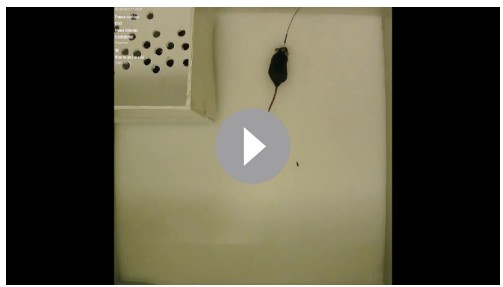

**Video 1.** Predator ambush. Example of a mouse fleeing after being ambushed by a remote-controlled mechanical spider hidden in an enclosure.
https://elifesciences.org/articles/90972/figures#video1

## SuM$^{VGLUT2+}$::POA neurons evoke active coping-like behaviors

To examine how activation of SuM$^{VGLUT2+}$::POA neurons contributes to responding to threatening stressors, we assessed behavioral changes evoked by photostimulation of SuM$^{VGLUT2+}$::POA neurons. We injected VGLUT2-Cre or WT (Cre- littermates) mice with Retro-AAV5-DIO-ChR2eYFP in the POA, and an optic fiber was placed over SuM (*Figure 4A*). We employed a paradigm with a 15-min trial divided into three 5 min periods: pre-stimulation, stimulation at 10 Hz, and post-stimulation. Based on review of videos obtained

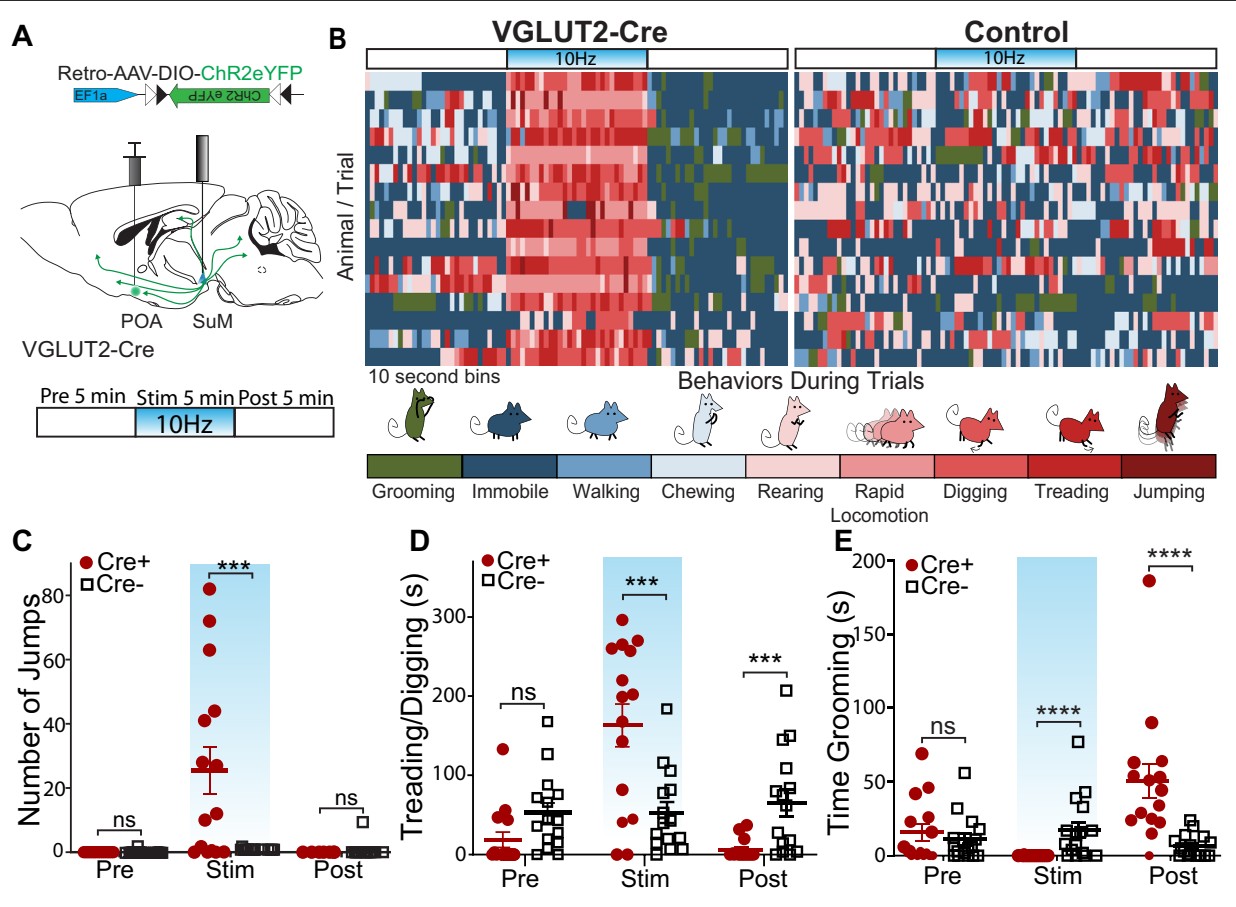

**Figure 4.** Photostimulation of SuM^VGLUT2+^::POA neurons evokes active coping behaviors. (**A**) Illustration of injections and fiber implant in VGLUT2 +Cre mice and schematic of photostimulation paradigm of 5 min pre, 5 min stimulation, and 5 min post. Behaviors were evaluated during each epoch. (**B**) For Cre+ (n=16) and Cre- (n=16) mice, the behavior in 10 s bins was scored based on the predominant behavior displayed during each 10 s period for the 15-min trial into grooming, stationary, walking, chewing, rearing, rapid locomotion, digging (moving bedding toward the tail), treading (move beading forward), and jumping. The graphic shows a by-animal scoring of the 15-min trial color-coded for each of the behavior categories. (**C**) During the stimulation period, Cre+ (n=15) mice show significantly (*p=0.012) greater jumps than Cre- (n=16) mice, and jumping behaviors were not significantly different during pre and post stim periods. (**D**) Behavior was also scored for time spent moving beading (digging or treading). Cre +mice showed significantly (**p=0.004) greater time engaging in digging/treading behaviors during the stimulation period, and significantly (*p=0.014) less time during the post-stimulation period compared to Cre- mice. (**E**) Behavior was also scored for time spent engaging in grooming behaviors, and Cre +mice spent significantly (*p=0.016) less time grooming during the stimulation period and significantly (**p=0.005) more time grooming during the post stimulation period compared to Cre- mice. All data plotted as mean ± SEM.

The online version of this article includes the following source data for figure 4:

**Source data 1.** Behaviors induced by photostimulation of SuM^VGLUT2+^::POA neurons.

during the 15-min trials, we observed behaviors that could be classified into nine distinct categories: grooming, immobile, walking, chewing of bedding, rearing, rapid locomotion (movement was limited by the size of the arena), digging (moving bedding toward the tail), treading (moving bedding forward with front paws), and jumping (*Figure 4B*). A blinded observer scored the behaviors in 10 s intervals for the predominant behavior observed during each interval, and each behavior was assigned a color for visualization. Color coded representation of trials from 16 Cre+ (n=16) and WT (n=16) mice is illustrated (*Figure 4B*). The behavioral pattern in the pre-stimulation period is similar between the Cre +and WT mice. During the stimulation period, a clear shift in behavior was evident in Cre +mice. Photostimulation induced rearing, treading, digging, rapid locomotion, and jumping. During the post-stimulation period, we observed a new pattern with Cre +mice spending time immobile and grooming. An example of a Cre+ animal is shown in *Video 2*.

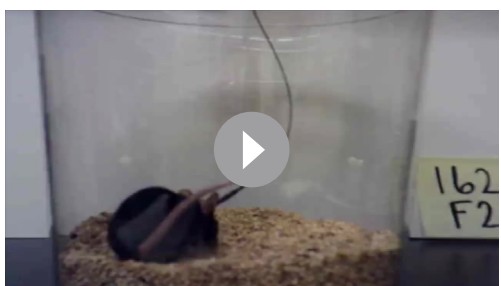

**Video 2.** Example of photostimulation of SuM<sup>VGLUT2+</sup>::POA neurons evoking active coping behaviors. Mice show an increase in active coping behaviors (e.g. jumping, digging) during photostimulation of SuM<sup>VGLUT2+</sup>::POA neurons.
https://elifesciences.org/articles/90972/figures#video2

We analyzed behavior, quantifying specific behaviors during each period. Quantification of jumping revealed that Cre +mice (n=16) engaged in significantly (p=0.012) more jumps during the stimulation period compared to WT (n=16) mice (*Figure 4C*). During the pre- and post-stimulation periods, there was not a significant difference in the number of jumps (pre-stimulation=p > 0.999 and post-stimulation=p > 0.999). We also observed that during the stimulation period the Cre+mice engaged in bouts treading/digging, vigorously moving bedding forwards and backwards, as previously described for defensive burying (*De Boer and Koolhaas, 2003*). Defensive burying, characterized by moving bedding forward or backwards often in alternating pattern, is evoked by threatening and noxious stimuli, and is a described active stress coping strategy in rodents (*De Boer and Koolhaas, 2003*; *Castro et al., 2016*; *Reynolds and Berridge, 2001*; *Richard and Berridge, 2011*). We conservatively quantified together the movement of bedding as treading/digging that may include spontaneous digging (*Figure 4D*) and found no significant (p=0.95) difference during the pre-stimulation period. In the stimulation period, there was a significant (p=0.004) increase in the time spent treading/digging, and, surprisingly, during the post-stimulation period time treading/digging was significantly (p=0.014) decreased in Cre +mice. The variability in behaviors can be attributed to exclusion of one behavior by the other, with individual mice having variability in the predominant behavior displayed during the photostimulation period, but all mice shifted to increased active coping behaviors. The behaviors evoked by photostimulation fit with active coping behaviors seen in the context of stressors. Specifically, escape (jumping, rapid locomotion), defensive burying/treading (digging, pushing beading forward).

We also quantified grooming behaviors in each of the three periods (*Figure 4E*). During the pre-stimulation period, there was not a significant difference (p>0.999). During the stimulation period, the Cre +mice did not engage in grooming, leading to a significant (p=0.016) decrease in time spent grooming compared to Cre- mice. In the post-stimulation period, Cre +mice showed a significant (p=0.005) increase in time spent grooming compared to Cre- mice. A reasonable interpretation of the rise in grooming post stimulation is that photostimulation evoked a stressed-like state and cessation of photostimulation led to selfcare grooming, as seen following acute stressors (*Kalueff and Tuohimaa, 2004*; *van Erp et al., 1994*). In summary, the analysis of behaviors elicited by photostimulation of SuM<sup>VGLUT2+</sup>::POA neurons without conditioned cues or concomitant stressors demonstrates a dramatic shift in behavior to escape oriented (jumping, rapid locomotion) and threat response behavior, including rearing and defensive burying, during the photostimulation period. In contrast to freezing, the behaviors elicited by activation of SuM<sup>VGLUT2+</sup>::POA neurons indicate that they may promote active coping strategies.

## Photostimulation of SuM<sup>VGLUT2+</sup>::POA neurons drives real-time avoidance

To examine whether activation of SuM<sup>VGLUT2+</sup>::POA neurons may contribute to the aversive aspects of threatening stress, we carried out real-time place aversion testing (RTPA) by pairing one side of the chamber with photostimulation of SuM<sup>VGLUT2+</sup>::POA (*Figure 5A*) neurons at multiple frequencies (1, 5, 10, and 20 Hz). Photostimulation of SuM<sup>VGLUT2+</sup>::POA neurons produced significant aversion at all frequencies in Cre+ (n=19) mice compared to Cre- (n=21) littermate control mice, and higher stimulation frequencies evoked greater aversion (p<0.001; *Figure 5B and C*). Example of Cre +mice with stimulation at 1, 5, and 10 Hz as well as Cre- mice is in *Video 3*. WT and Cre +mice explored equivalently, with similar number of entries to the stim side (p=0.41; *Figure 5D*). Cre +mice quickly left the stimulation side, and the average time spent on the stimulation side of the area was significantly lower (p<0.001; *Figure 5E*). These data indicate, surprisingly, that photostimulation, although aversive, did

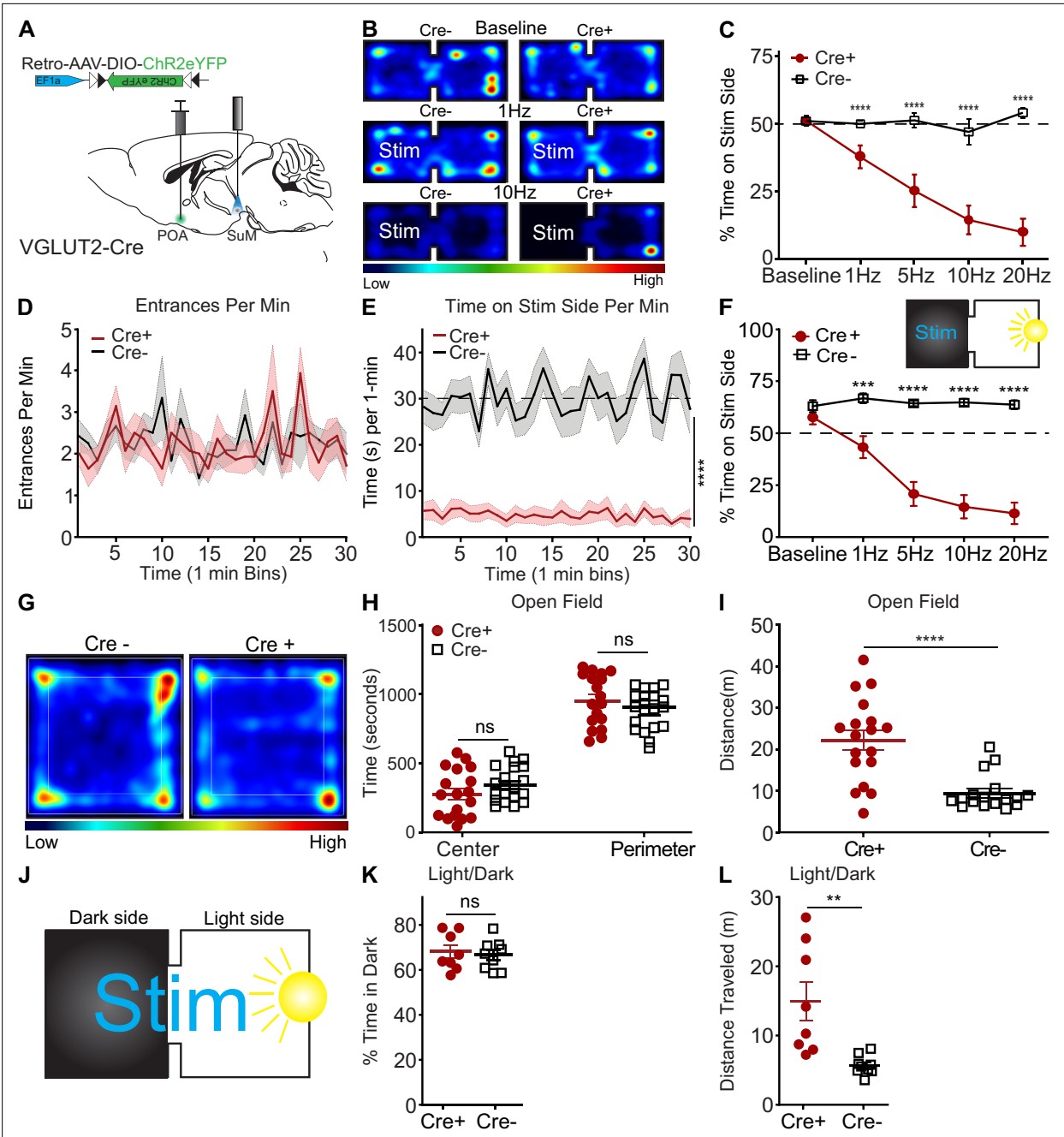

**Figure 5.** Activation of SuM^VGLUT2+^::POA neurons drives aversion but does not promote anxiety-like behavior. (**A**) Schematic of injection of Retro-AAV-ChR2eYFP into the POA and placement of a midline optic fiber over SuM. (**B**) Representative heat maps for Cre +and Cre- mice at baseline (no stim) and 10 Hz photostimulation in two-sided arena with photostimulation associated with one side. (**C**) Quantification of time spent on stimulation side for Cre+ (n=19) and control Cre- (n=21) mice showing frequency-dependent increase in avoidance of the stimulation side of the area (***p<0.001, ****p<0.0001). (**D**) Mean ± SEM entrances per min (in 1 min bins) to the during 10 Hz stimulation trials were not significantly (p=0.87) different between Cre+ (n=14) and Cre- (n=13) mice. (**E**) Mean time spent on the stimulation side for Cre- and Cre +in 1 min bins during 10 Hz stimulation trials was significantly (p<0.0001) lower in Cre +mice. (**F**) (Inset) Diagram of light/dark arena with photostimulation provided on the dark side of the arena and quantification of time spent on dark (stimulation) side, demonstrating that Cre- (n=12) animals show a baseline preference for the dark side of arena that is overcome by photostimulation with Cre+ (n=13) mice spending significantly (**p=0.002, ***p<0.001, ****p<0.0001) less time on the stimulation side. (**G**) Representative heat map of Cre- and Cre +during 10 Hz photostimulation in open field test. (**H**) In the open field test, time spent in center and perimeter were not significantly different (p=0.19) between Cre+ (n=18) and Cre- (n=17) during 10 Hz photostimulation. (**I**) Distance traveled during open field testing was significantly (p<0.001) increased in Cre +compared to Cre- mice. (**J**) Schematic of real time light/dark choice testing with stimulation provided at 10 Hz throughout the arena. (**K**) Both Cre+ (n=8) and Cre- (n=9) mice showed a preference for the dark portion of the arena but

*Figure 5 continued on next page*

*Figure 5 continued*

were not significantly (p=0.9) different. (**L**) The total distance traveled by Cre +mice were significantly (p=0.003) greater than Cre- mice. All data plotted as mean ± SEM.

The online version of this article includes the following source data and figure supplement(s) for figure 5:

**Source data 1.** Photostimulation of SuM$^{VGLUT2+}$::POA neurons drive aversion.

**Figure supplement 1.** GABAergic neurons in SuM do not drive place preference.

**Figure supplement 1—source data 1.** Effects of photostimulation of GABAergic neurons in SuM.

not generate aversive pairing, as found for other brain areas (*Kravitz et al., 2012*). To examine the relative aversiveness of SuM$^{VGLUT2+}$::POA photostimulation, we carried out RTPA experiments using an arena with a dark side and a bright side (*Figure 5F* inset). We paired photostimulation with the dark side. As expected, Cre- mice show a preference for the dark side of the arena, however, photostimulation of the SuM$^{VGLUT2+}$::POA neurons yielded significant aversion of Cre+ (n=13) compared to Cre- (n=12) (p<0.001; *Figure 5F*). Cre +mice spent nearly the entire trial in the brightly lit side, demonstrating photostimulation of SuM$^{VGLUT2+}$::POA neurons drove avoidance sufficient to overcome a mildly aversive stimulus (bright light).

Reports have described multiple populations of neurons in supramammillary including populations that project to the dentate gyrus of the hippocampus that release both GABA and glutamate. Our anatomic studies indicated that SuM$^{VGLUT2+}$::POA neurons do not project to the dentate gyrus (*Figure 2*) and are not GABAergic (*Figure 1—figure supplement 1G–L*). Thus, SuM$^{VGLUT2+}$::POA neurons are not part of either a GABAergic or dual transmitter population. To examine if GABAergic neurons in SuM (SuM$^{VGAT+}$) can mediate real-time place aversion or preference, we injected VGAT-Cre mice with AAV to express ChR2 (n=26) or control (n=26) in SuM and carried out real-time place aversion testing. Photostimulation at 10 Hz yielded no significant aversion or preference compared to baseline (p=0.838) (*Figure 5—figure supplement 1A–D*). This is in contrast to the robust aversion caused by photostimulation of SuM$^{VGLUT2+}$::POA neurons.

## Photoactivation of SuM$^{VGLUT2+}$::POA neurons does not cause anxiogenic-like behavior

Threat can induce anxiety-like, risk aversion behavioral states (*McCall et al., 2015*; *Felix-Ortiz et al., 2013*; *Jakovcevski et al., 2011*). To test whether activity of SuM$^{VGLUT2+}$::POA neurons contributes to anxiety-like behaviors, we used two established assays for anxiety-like behavior: open field and light-dark exploration (*Bailey, 2009*; *Heredia et al., 2014*; *Prut and Belzung, 2003*; *Chaouloff et al., 1997*). During open field testing, 10 Hz photostimulation was applied to SuM$^{VGLUT2+}$::POA neurons. We quantified the time spent in the perimeter (outer 50%) vs. the center (inner 50%) of the arena and found VGLUT2-Cre+ (n=18) and WT (n=17) mice did not significantly differ (center p=0.17 and perimeter p=0.19; *Figure 5G and H*). The total distance traveled was significantly (p<0.001) increased in in Cre +mice compared to WT mice (*Figure 5I*). Increased distance traveled in some mice was not surprising, because prior testing (*Figure 4*) revealed the mice engaged in escape behaviors during photostimulation. We observed behaviors including jumping during open field testing as well. We also carried out open field testing using VGAT-Cre mice to selectively activate SuM$^{VGAT+}$ neurons and observed no significant effect on time in center (p=0.254) or perimeter (p=0.287) (*Figure 5—figure supplement 1F*), but mice expressing ChR2eYFP displayed a small but significant (p=0.035) decrease in distance traveled (*Figure 5—figure supplement 1G*). We tested if photostimulation of SuM$^{VGLUT2+}$::POA neurons would increase preference for the dark area, a potential sign of elevated anxiety (*Figure 5J*). We

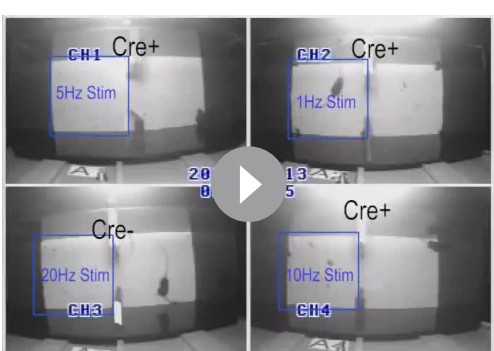

**Video 3.** Example of photostimulation of SuM$^{VGLUT2+}$::POA neurons driving aversion. Mice avoid the photostimulation side with higher frequencies showing greater aversion.

https://elifesciences.org/articles/90972/figures#video3

found that photostimulation of SuM$^{VGLUT2+}$::POA neurons did not significantly alter preference (p=0.68; *Figure 5K*) in VGLUT2-Cre+mice (n=8) compared to WT (n=9). The activation of SuM$^{VGLUT2+}$::POA neurons evoked escape behaviors in open field testing, as seen in *Figure 4*; similarly, a significant increase in total distance was detected during light-dark exploration assay in Cre +compared to WT (p=0.003; *Figure 5L*). The findings we obtained in open field and light-dark exploration testing do not support a role for SuM$^{VGLUT2+}$::POA neurons in driving anxiety-like behaviors.

## SuM$^{VGLUT2+}$::POA neurons can drive instrumental action-outcome behavior

Current theoretical frameworks for examining threat responses divide behaviors into two broad categories: innate (fixed) and instrumental (action-outcome; *LeDoux and Daw, 2018*). Additionally, stress responses can be divided into active vs passive actions. Behaviors in these separable categories are mediated by distinct neural circuits (*LeDoux and Daw, 2018*; *Fadok et al., 2017*). For example, areas involved in responding to specific threatening stimuli (looming threat) drive an innate fixed behavior repertoire (*LeDoux and Gorman, 2001*). Photostimulation of SuM$^{VGLUT2+}$::POA neurons evoked multiple active-like behaviors in response to threatening or noxious stimuli (*Figure 4*). To test if SuM$^{VGLUT2+}$::POA neurons can promote flexible repertoires of behavior including instrumental tasks, as opposed to only innate behaviors (ex. fleeing), we used an operant negative reinforcement paradigm (*Figure 6A–B*). Because our results show photostimulation of SuM$^{VGLUT2+}$::POA neurons is aversive, photostimulation of SuM$^{VGLUT2+}$::POA neurons could be used as a negative reinforcer in a negative reinforcement paradigm. We employed a paradigm using two nose poke ports, one active and one inactive. SuM$^{VGLUT2+}$::POA neurons were photostimulated at 10 Hz during a 10-min trial. Activation of the active port by a nose poke triggered a 10 s pause in photostimulation of SuM$^{VGLUT2+}$::POA neurons and turned on a house light for 10 s (*Figure 6A and B*). Following this 10 s pause, the house light switched off and photostimulation resumed until the subsequent activation of the active port. We tested animals for 4 consecutive days with the 10 min trials without prior training. Importantly, performance of the task had to be completed during photostimulation.

We hypothesized that if stimulation of SuM$^{VGLUT2+}$::POA evokes fixed innate defensive behaviors, or simply promotes high amounts of locomotion, then performance of an operant task (an action-outcome behavior) with active and inactive ports would be impaired because innate defensive behaviors would conflict with performance of the operant task. Importantly, results obtained can help differentiate general elevation in locomotor activity from directed coping behavior. Analyses of port activations during the 10-min trials revealed that Cre +mice (n=11) paused the photostimulation (reward) significantly more on all 4 days of testing compared to Cre- littermate (n=11) mice (Interaction$_{Genotype \times Day}$ p<0.001; Day 1 p<0.001; Day 2 p<0.001; Day 3 p<0.001; Day 4 p<0.001; *Figure 6C*). The number of 10 s pauses approached the maximum, 60 possible in a 10 min period. Cre +mice engaged the active port significantly more than Cre- mice (p<0.001; *Figure 6D*) and this effect was consistent for every training session (Day 1 p<0.0001 = ; Day 2 p<0.0001; Day 3 p<0.0001; Day 4 p=0.0011). There was a slight difference in baseline performance of the task. The time to the first engagement of the active port was significantly different for Cre +compared to Cre- mice during the first test trial (p=0.046). During the three subsequent tests, Cre +mice were significantly faster to engage the active port after the start of the trial compared to Cre- mice (Day 2 p=0.009; Day 3 p=0.003; Day 4 p=0.034; *Figure 6E*). Cre +mice engaged the active port significantly more than the inactive port on all test days (p<0.001; *Figure 6—figure supplement 1A*). There was not a significant difference in Cre +and Cre- mice for engagement of the inactive port on any day (Cre +p = 0.30; Cre- p=0.055; *Figure 6—figure supplement 1A–B*).Furthermore, Cre +mice were significantly faster in the second through fourth trials at engaging the active port compared the first trial (Day 1 vs Day 2 p=0.034; vs Day 3 p=0.021; vs Day 4 p=0.027; *Figure 6E*). Suggesting that mice became more proficient during subsequent trials.

To examine effort-related motivation, we employed a 30-min test using a progressive ratio requiring an exponentially increasing number of active port activations to trigger a pause in the photostimulation on the fifth day of operant behavior testing (*Figure 6B*; *Parker et al., 2019*). During trials using a progressive ratio, Cre +mice activated the active port and paused stimulation significantly (p<0.001) more times than Cre- mice (*Figure 6F*). Examination of individual cumulative active port activation data showed Cre +mice continued to engage the active port throughout progressive ratio trials

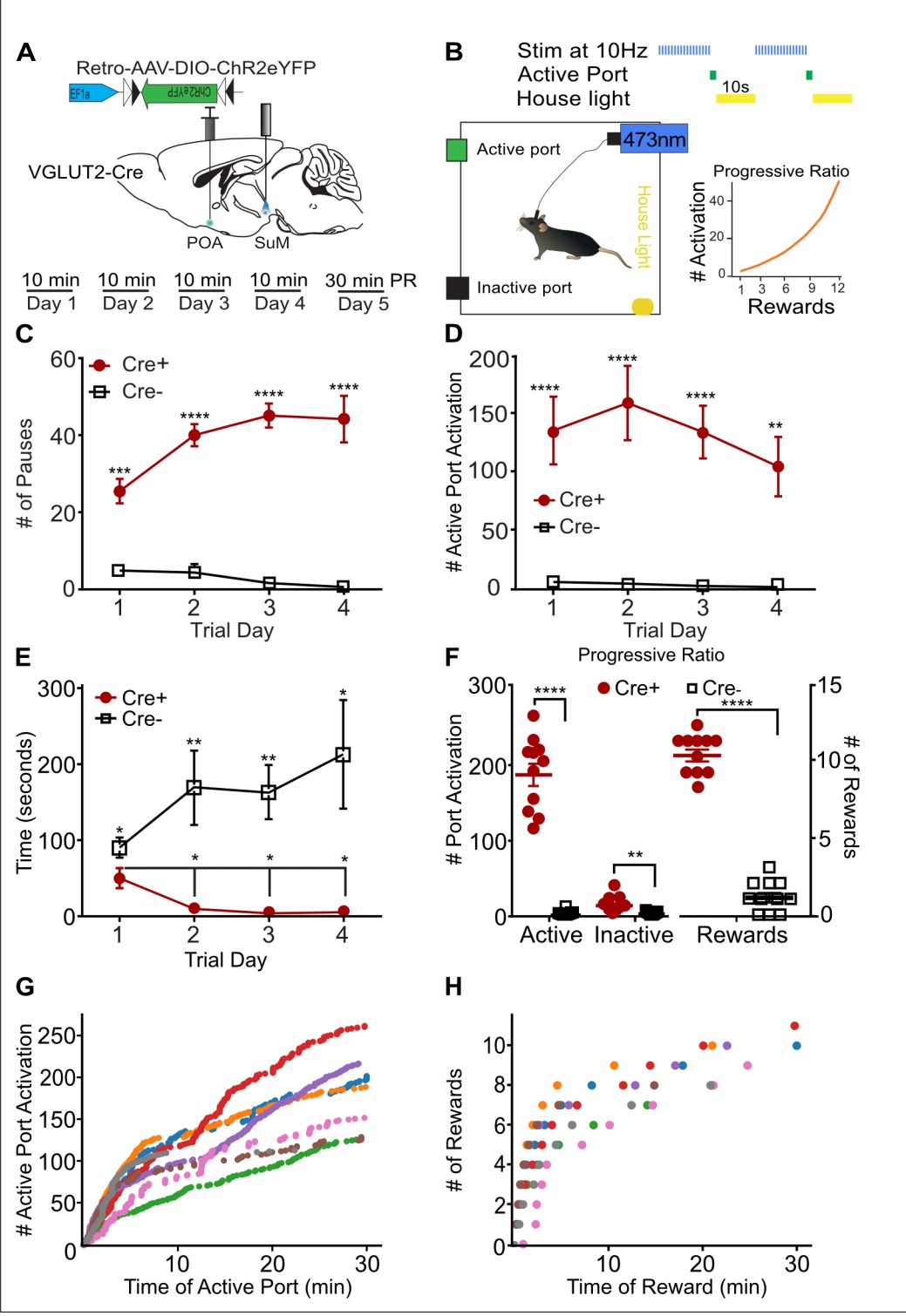

**Figure 6.** SuM$^{VGLUT2+}$::POA neurons can drive instrumental action-outcome operant behavior. (**A**) Schematic of injection and implant in VGLUT2 +Cre mice and paradigm of testing in 10 min trials with 4 days before a progressive ratio (PR) trial on day 5. (**B**) Illustration of the testing paradigm and set up. 10 Hz photostimulation was applied during the trials. Activation of the active port triggered the house light and paused stimulation for 10 s. Also shown is the progressive ratio used with number of required port activations per reward on vertical axis and reward number on the horizontal. (**C**) Cre +mice (n=11) activated the active port triggering significantly (****p<0.0001) more pauses in stimulation than Cre- mice (n=11) mice on all 4 days of testing. (**D**) Cre +mice

*Figure 6 continued on next page*

*Figure 6 continued*

activated the port significantly (****p<0.0001, ***p<0.001) more than the Cre- mice on all four trials. (**E**) The time to first activation of the active port was significantly (*p=0.046) different during the first trial, and Cre +mice activated the active port significantly (**p<0.01, **p<0.01, **p=0.034) faster on subsequent trials. (**F**) On the fifth day after four 10-min trials, mice were tested for 30 min using a progressive ratio. Cre +performed significantly (****p<0.001) more active port activations than Cre- mice and activated the inactive port significantly more (**p=0.001) times. Cre +mice also triggered significantly (****p<0.001) more pauses in photostimulation than Cre- mice. (**G**) Individual data for a representative (n=7) cohort of Cre +mice showing cumulative active port activations as a function of time during the progressive ratio test show on-going engagement of the active port throughout the 30 min trial. (**H**) The cumulative pauses in photostimulation (rewards) earned as a function of time during the progressive ratio trial are shown for individual Cre +mice. Mice earned between 7 and 11 pause rewards during the trial. All data plotted as mean ± SEM.

The online version of this article includes the following source data and figure supplement(s) for figure 6:

**Source data 1.** Photostimulation of SuM$^{VGLUT2+}$::POA neurons can drive instrumental action-outcome operant behavior.

**Figure supplement 1.** SuM$^{VGLUT2+}$::POA stimulation drives active port activation.

**Figure supplement 1—source data 1.** Number of activate vs inactive port activations in Cre +and Cre- mice.

despite the increasing work required to generate each pause in photostimulation. No breakpoint was observed during the trial (*Figure 6G and H*). This finding indicates that activation of SuM$^{VGLUT2+}$::POA neurons remains salient and motivating through the tested period and that photostimulation of SuM$^{VGLUT2+}$::POA does not evoke behaviors precluding completion of the task. Examination of reward behavioral epochs for representative animals shows that for the lowest total number of rewards earned (green dots), the next pause required 20 activations of the active port, and for the highest (red dots), the next pause required 50 activations of the active port. Taken together, results from the instrumental reinforcement tasks demonstrate that activation of SuM$^{VGLUT2+}$::POA neurons does not solely evoke innate stereotyped behaviors and can drive active coping in the form of instrumental behaviors.

## SuM$^{VGLUT2+}$::POA neurons are recruited during active coping behaviors during forced swim

The forced swim test is increasingly established as an assay of coping strategy with passive and active components, and we used forced swim testing in conjunction with fiber photometry to examine if SuM$^{VGLUT2+}$::POA neurons are recruited differentially during active (mobile) or passive (immobile) behaviors (*Costa et al., 2013*; *Commons et al., 2017*). During a first 15-min trial of forced swimming, mice shifted from active escape behaviors (wall climbing, robust swimming) to passive (immobile floating) behaviors. We carried out experiments using fiber photometry paired with automated computer-based behavior analysis (*Hu et al., 2023*). The automated behavior scoring enabled high temporal resolution analysis of behaviors (*Video 4*) and each frame of the video was coded for the classified behavior and plotted as a color-coded point. We observed four types of behavior during forced swimming: climbing, swimming, hindpaw swimming, and immobile floating, and we trained a model to identify each. Analysis of behavior showed that after the first minutes of the trial, mice shifted from swimming and climbing to immobile floating with intermittent swinging or hindpaw swimming. We analyzed fiber photometry recordings to examine whether recruitment of SuM$^{VGLUT2+}$::POA neurons changed with this shift to more passive behaviors. As shown in the representative trace (*Figure 7A*), the Ca$^{2+}$-dependent signal, but not the isosbestic signal, markedly decreased at the time the shift in behavior was

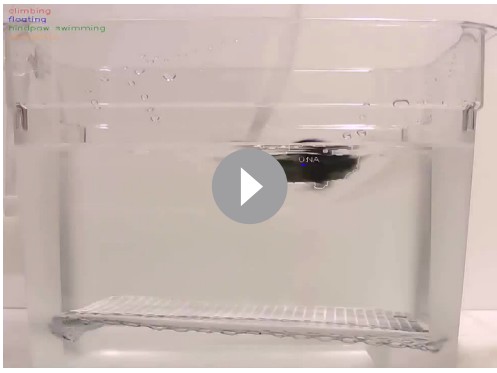

**Video 4.** Annotated video showing color-coded behaviors with concurrent calcium activity recording during forced swim test. Climbing is shown in red, floating in blue, hindpaw swimming in green, and swimming in yellow.

https://elifesciences.org/articles/90972/figures#video4

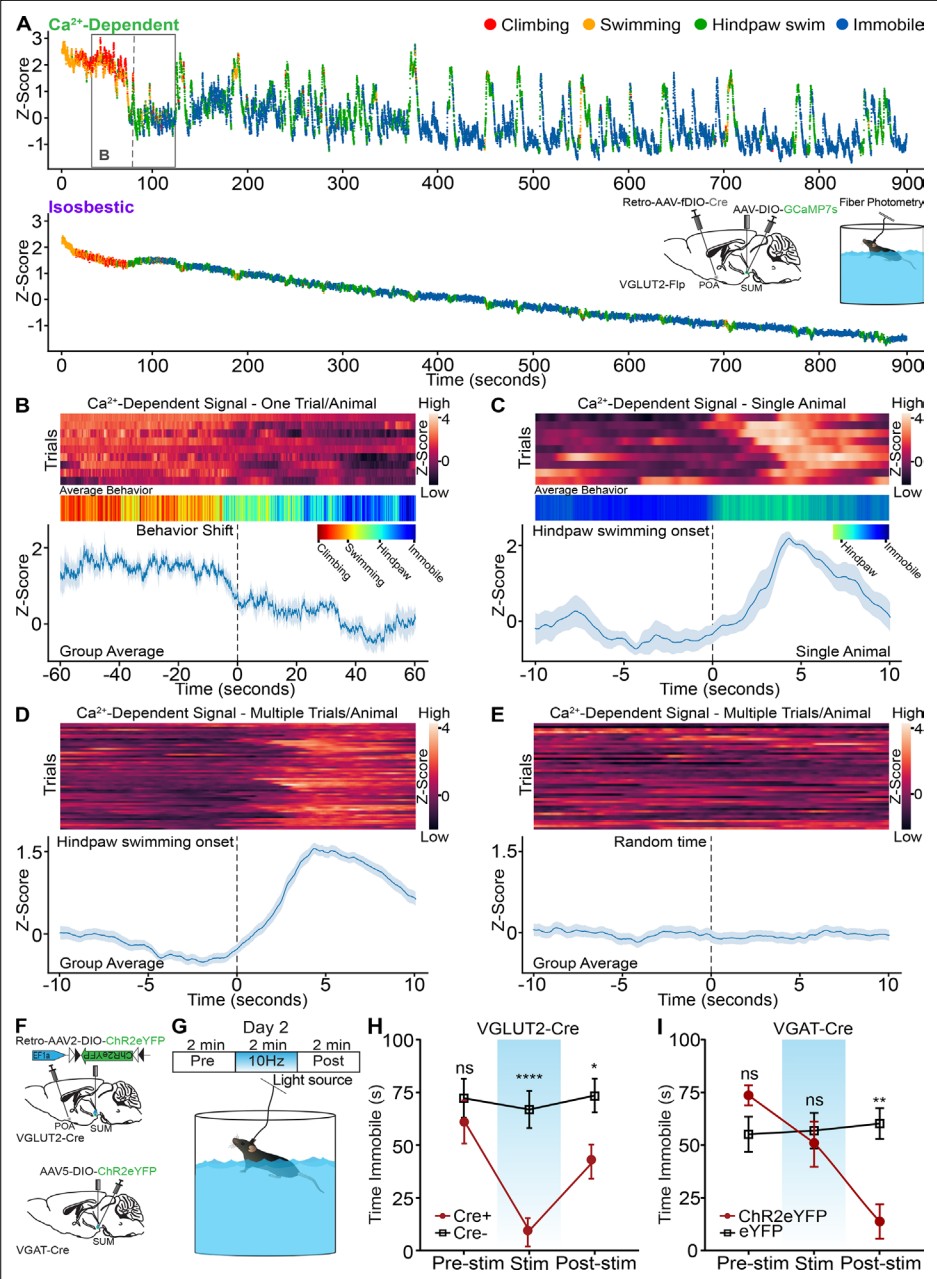

**Figure 7.** SuM$^{VGLUT2+}$::POA neurons are recruited during active coping, and activation is sufficient drive a switch to active coping. (**A**) Color-coded per video frame (dot color) behavior combined with normalized Ca$^{2+}$-dependent and isosbestic fiber photometry signals. Scoring and analysis of climbing, swimming, hindpaw swimming, and immobility were completed using deep learning-based classification and quantification (LabGym) of recorded behavior. Inset shows injection, implants, and behavioral assay. Box highlights the time frame shown in B. (**B**) Heat maps for recordings obtained from mice (n=9) during forced swim, averaged color-coded behaviors, and mean ± SEM Z-score of Ca$^{2+}$-dependent signal for 2-min period around behavioral shift from climbing and swimming to hindpaw swimming and immobility, showing significant (*95% CI) decline in the Ca$^{2+}$-dependent signal. (**C**) Representative heat map, averaged behavior, and mean ± SEM Z-score of Ca$^{2+}$-dependent signal for 20 s window around onset of hindpaw swim from a representative animal. (**D**) Representative heat map and mean ± SEM Z-score of Ca$^{2+}$-dependent signal for 20 s window around onset of hindpaw swim from for events from nine mice. Engaging in hindpaw swim correlates with a brief, significant (***99.9% CI) increase in Ca$^{2+}$ signal. (**E**) Representative heat map and mean ± SEM Z-score of Ca$^{2+}$-dependent signal for 20 s window around random time points. (**F**) Schematic of injection and implant in VGLUT2-Cre or VGAT-Cre mice. (**G**) Illustration of testing paradigm on second forced swim test following 15 min swim on first day. (**H**) The average time spent immobile

*Figure 7 continued on next page*

*Figure 7 continued*

during the pre-stimulation period was not significantly (p>0.999) different. During 10 Hz photostimulation VGLUT2-Cre+mice engaged in vigorous swimming and the time spent immobile was significantly (****=p < 0.001) less than Cre- mice. In the post-stimulation period, the time spent immobile remained significantly (*=0.047) lower in Cre+mice compared to Cre-. (**I**) In VGAT-Cre + mice expressing ChR2 (n=11), the average time spent immobile during the pre-stimulation period was not significantly (p=0.218) different compared to eYFP controls. Photostimulation did not significantly increase time immobile (p>0.999). However, there was a significant (**p=0.001) decrease in the time spent immobile compared to the post-stimulation period. For $Ca^{2+}$ signal differences: *=95% CI, **=99% CI, ***=99.9% CI, ns = not significant. All data plotted as mean ± SEM unless otherwise noted.

The online version of this article includes the following source data for figure 7:

**Source data 1.** Photometry data for Figure A-E.

**Source data 2.** Data for photostimulation of SuM VGLUT2 and VGAT neurons during FST.

occurring. To combine behavioral data, we assigned numerical values, one through four, to each behavior: climbing (4), swimming (3), hindpaw swimming (2), and immobile floating (1). The mean was plotted as a heat map (red higher values to blue lower values) to be compared to time locked fiber photometry data (*Figure 7B*). Examining the timeframe around the behavioral shift in each animal, we found that as behavior shifted from active coping (climbing, swimming) to more passive strategies and greater immobility, the $Ca^{2+}$ signal significantly (95% CI) declined (*Figure 7B*). Examining the data after this initial transition, we observed that many elevations in the Z-score were accompanied by changes in behavior to swimming or hindpaw swimming (*Figure 7A*). To further examine this possible correlation, we examined the time frame around transition to hindpaw swimming for changes in $Ca^{2+}$ signal. In a 20-s window centered on the onset of hindpaw swimming, we found that hindpaw swimming was accompanied by a rise in the $Ca^{2+}$-dependent signal across multiple events within a trial (*Figure 7C*). A similar analysis across events in multiple animals demonstrated a significant (99.9% CI) rise in the $Ca^{2+}$-dependent signal from $SuM^{VGLUT2+}$::POA neurons during the shift from immobility to hindpaw swimming or swimming behaviors (*Figure 7D*). Time series analysis of 20 s epochs centered on random time intervals did not reveal any rise in the $Ca^{2+}$-dependent signal (*Figure 7E*). These data indicate that recruitment of $SuM^{VGLUT2+}$::POA neurons fluctuates with changes in coping strategy during a forced swim assay with decreased engagement of this circuit during times of immobility.

## $SuM^{VGLUT2+}$::POA neurons promote active coping during forced swim test

We found that $SuM^{VGLUT2+}$::POA neurons are activated by acute stressors (*Figure 3*) and are recruited during times of greater active coping behaviors (*Figure 7*). To test if $SuM^{VGLUT2+}$::POA neurons can drive a switch in coping strategy in the context of ongoing stressors, we used a two-day forced swim stress test (*Commons et al., 2017*; *Seo et al., 2019*). We tested animals during the second day, when immobile floating is the predominant behavior. As in previous experiments, we used VGLUT2-Cre (Cre+) or WT (Cre-) mice injected in the POA bilaterally with Retro-AVV-DIO-ChR2eYFP. On day 1, we subjected mice to 15 min of forced swim. On the subsequent day, we repeated the forced swim for 6 min divided into three periods: pre-stimulation, stimulation at 10 Hz, and post-stimulation (*Figure 7F*). Trials on the second day were recorded and scored by a blinded observer for time spent immobile in each 2 min period. An example of a Cre +mouse with stimulation 10 Hz is shown in *Video 5*. During the pre-stimulation period, there was not a significantly (p>0.999) different amount of time immobile between Cre+ (n=9) and Cre- (n=11) mice. During the stimulation period, Cre +mice began swimming and attempting to climb the wall of the circular swim arena, reflected by significantly (p<0.001) less time immobile (*Figure 7H*). Interestingly, in the post stimulation phase, the difference in time spent immobile between Cre +can WT mice decreased but remained

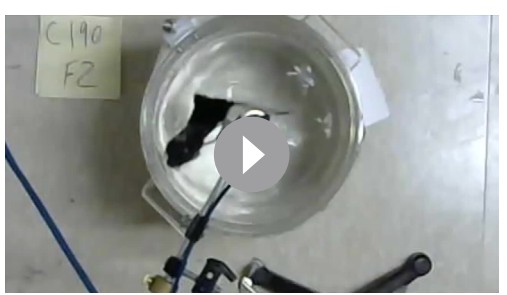

**Video 5.** Photostimulation of $SuM^{VGLUT2+}$::POA neurons during forced swim test elicits active coping behaviors. https://elifesciences.org/articles/90972/figures#video5

significantly (p=0.047) different. These results indicate that in the context of an ongoing stressor, activation of SuM$^{VGLUT2+}$::POA neurons is sufficient to trigger a change in coping strategy from largely passive (floating) to active (swimming, climbing). The persistent effect of the acute activation into the poststimulation phase suggests that activation of SuM$^{VGLUT2+}$::POA neurons may shift how the stressor is processed or approached.

We next examined the effect of 10 Hz photostimulation of SuM$^{VGAT}$ neurons on coping strategy using the same 2-day swim paradigm. VGAT-Cre mice were injected in the SuM to express ChR2eYFP (n=10) or control (eYFP) (n=11), and a fiber optic was placed over SuM (*Figure 7F*). Quantification of behavior revealed no significant difference in time spent immobile during the pre-stimulation (p=0.218) or the stimulation (p>0.999) periods. Surprisingly, in the post-stimulation period, we observed a dramatic shift in behavior, marked by a significant (p=0.047) decrease in time spent immobile (*Figure 7I*). The amplitude of change in behavior was similar to what we observed during the stimulation phase of the experiments on SuM$^{VGLUT2+}$::POA neurons (*Figure 7H*). One interpretation of these data is that release of sustained local inhibition leads to rebound activity of output SuM$^{VGLUT2+}$::POA neurons.

## Suppression of SuM$^{VGLUT2+}$::POA neurons is required for feeding

Feeding and responding to threats are conflicting actions, and SuM$^{VGLUT2+}$::POA neurons may play a role in switching between behavioral paradigms (e.g. feeding vs escape). We sought to examine SuM$^{VGLUT2+}$::POA neuron activity in relation to feeding using fiber-photometry-based GCaMP recordings. To promote differential drive for food, mice were given ad lib access to food (fed condition) or food deprivation for 24 hours, and on testing day, were presented with a chow pellet, while Ca$^{2+}$-dependent GCaMP fluorescence and isosbestic signals were recorded. In the fed state, mice spent significantly less time interacting with the food (p<0.001; *Figure 8C*) and ate significantly less (p<0.001; *Figure 8D*) compared to food deprived state. We analyzed the Ca$^{2+}$-dependent and isosbestic signals around presentation of food to fed and food deprived mice. In the food deprived state, mice spent more time with the food and ate more food (*Figure 8E and F*). At the time of food presentation, the Ca$^{2+}$-dependent signal decreased significantly (99.9% CI) but not the isosbestic signal. The change in the Ca$^{2+}$-dependent signal was larger and more sustained when the animals spent longer interacting with the food (*Figure 8E and F*). Together, the data support suppression of SuM$^{VGLUT2+}$::POA neural activity during consummatory behavior with greater consumption associated with enhanced suppression.

To examine whether activation of SuM$^{VGLUT2+}$::POA neurons disrupted consummatory behavior, we tested the impact of photostimulation on feeding behavior in food deprived mice. Using VGLUT2-Cre (Cre+) or WT (Cre- littermates) mice injected with Retro-AAV2-DIO-ChR2eYFP into the POA and implanted with a fiber optic midline over SuM, we examined how feeding behaviors in food deprived mice were altered by 10 Hz photostimulation of SuM$^{VGLUT2+}$::POA neurons (*Figure 8G–I*). We used trials lasting 20 min with unrestricted access to food added at the start of the trial. Each trial was divided into three periods: a 5 min pre-simulation period, 10 min stimulation period, and a 5 min post-stimulation period. Trials were recorded and scored for time spent interacting with the food pellet. During the pre-stimulation period, Cre+ (n=16) and Cre- (n=17) mice spent interacting with the food was not significantly different (p=0.47). Upon the introduction of photostimulation, the Cre +mice stopped eating and interacting with the food. During the photostimulation period, Cre +mice spent significantly less time interacting with the food compared to Cre- mice (p<0.001). Surprisingly, the decrease in time spent interacting with food continued into the post-stimulation phase, where Cre +mice continued to interact with the food significantly less than Cre- mice (p<0.001; *Figure 8H*). Reflecting the decrease in time spent with the food, Cre +mice consumed significantly less food during the 20 min trial than Cre- mice (p=0.001; *Figure 8I*). These results indicate that SuM$^{VGLUT2+}$::POA neurons can redirect behavior away from consumption, and suppression of SuM$^{VGLUT2+}$::POA neurons is required for feeding, even in a food deprived state. Ethologically, animals must choose rapidly between responding to threats and feeding when foraging. The findings here implicate this pathway in control of switching between these behaviors.

## Discussion

Together, the presented data show SuM$^{VGLUT2+}$::POA neurons function as a hub, with connections to many brain areas, to regulate responses to threatening stressors (*Figure 9*). We report here that

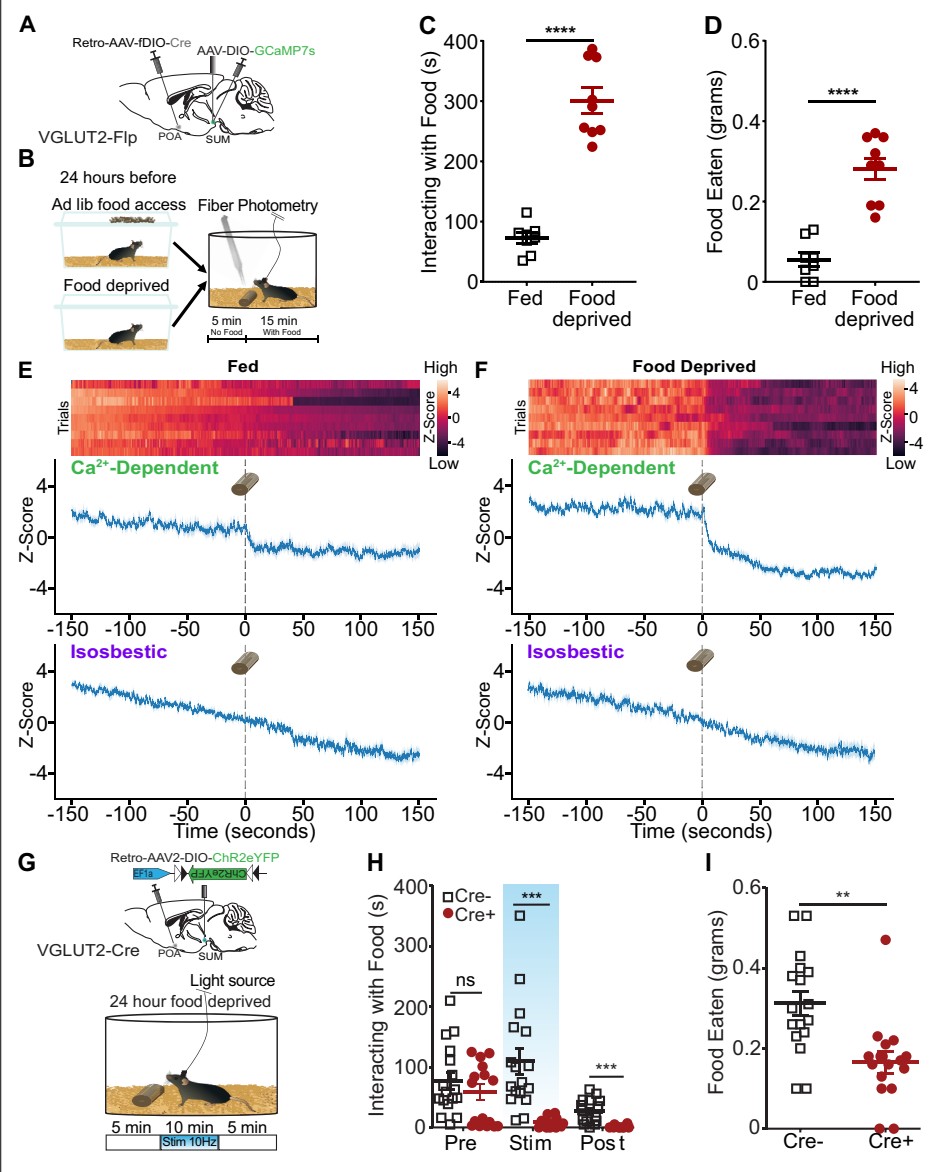

**Figure 8.** Consummatory behavior suppresses SuM^VGLUT2+^::POA neuron activity, and SuM^VGLUT2+^::POA neuron blocks food consumption. (**A**) Schematic of injections in VGLUT2-Flp mice with AAV-DIO-GCaMP7s in SuM and Retro-AAV-fDIO-Cre in the POA and a fiber placed over SuM. (**B**) Mice were given ad lib food access or food deprived for 24 hr prior to testing. After being placed in the arena and allowed to habituate, mice were given a 20 min trial divided into 5 min of baseline and 15 min after a chow pellet was added to the arena. (**C**) Food deprived mice (n=9) spent significantly (p<0.001) more time interacting with the food pellet and (**D**) ate significantly (p<0.001) more than in the fed state. (**E**) Heat map and mean ± SEM Z-scores for Ca^2+^-dependent and isosbestic signals for recordings obtained from 9 animals show a small drop in (**E**) fed mice state (*95% CI) compared to a sharp decrease in Ca^2+^ activity after the introduction of the food pellet following addition in (**F**) food deprivation (***99.9% CI). (**G**) Mice were fasted for 24 hr prior to testing, and the schematized 20 min paradigm was used. Following food deprivation, animals were given access to a chow pellet. The trial was divided into 5 min pre- and post-stimulation periods, with a 10 min period of stimulation. (**H**) The time spent interacting with the food was quantified for each period. Cre+ (n=16) and Cre- (n=17) mice both rapidly engaged with the food pellet, and the average time spent interacting was not significantly (p=0.47) different. During the stimulation period and the post-stimulation period, the average time interacting with the food was significantly (p=0.0006, p=0.0002) lower in Cre +mice compared to Cre- mice. (**I**) The food eaten during the total trial was calculated based on pellet weights, and, on average, Cre+ (n=16) mice ate significantly (p=0.001) less food during the trial than Cre- mice (n=17). (**p=0.001, ***p<0.001, ****p<0.0001). For Ca^2+^ signal differences: *=95% CI, **=99% CI, ***=99.9% CI, ns = not significant. All data plotted as mean ± SEM unless otherwise noted.

*Figure 8 continued on next page*

*Figure 8 continued*
The online version of this article includes the following source data for figure 8:

**Source data 1.** Food interaction and food eaten data for *Figure 8C and D*.

**Source data 2.** Photometry data for *Figure 8E and F*.

**Source data 3.** Food interaction and food eaten data for *Figure 8H and I*.

SuM^VGLUT2+^::POA neurons have broad arborizations to multiple brain regions involved in responding to threat and stress (*Figures 1 and 2*, and *Figure 1—figure supplement 2*). We identified diverse threatening stressors (dunk, shock, and predator), which elicit differential behavioral responses, swimming, jumping, and fleeing, recruit SuM^VGLUT2+^::POA neurons. Using a forced swim test, we found that recruitment of SuM^VGLUT2+^::POA neurons is correlated with periods of spontaneous active coping (swimming, climbing) and not with passive (immobile) behaviors. These data collectively indicate that SuM^VGLUT2+^::POA neurons are recruited by threatening stressors, preferentially during active coping. We also examined the roles of SuM^VGLUT2+^::POA neurons in responding to stressors by stimulating SuM^VGLUT2+^::POA neurons, and selective activation of SuM^VGLUT2+^::POA neurons evoked active coping-like behaviors in the absence of a stressor (*Figure 4*). SuM^VGLUT2+^::POA neuron activation drove aversion but did not promote anxiety-like behaviors (*Figure 5*). We also used a negative reinforcement test, requiring selective activation of an active port over the inactive port to show SuM^VGLUT2+^::POA neurons can drive flexible, in contrast to fixed or reflexive, behaviors (*Figure 6*). Activation of SuM^VGLUT2+^::POA

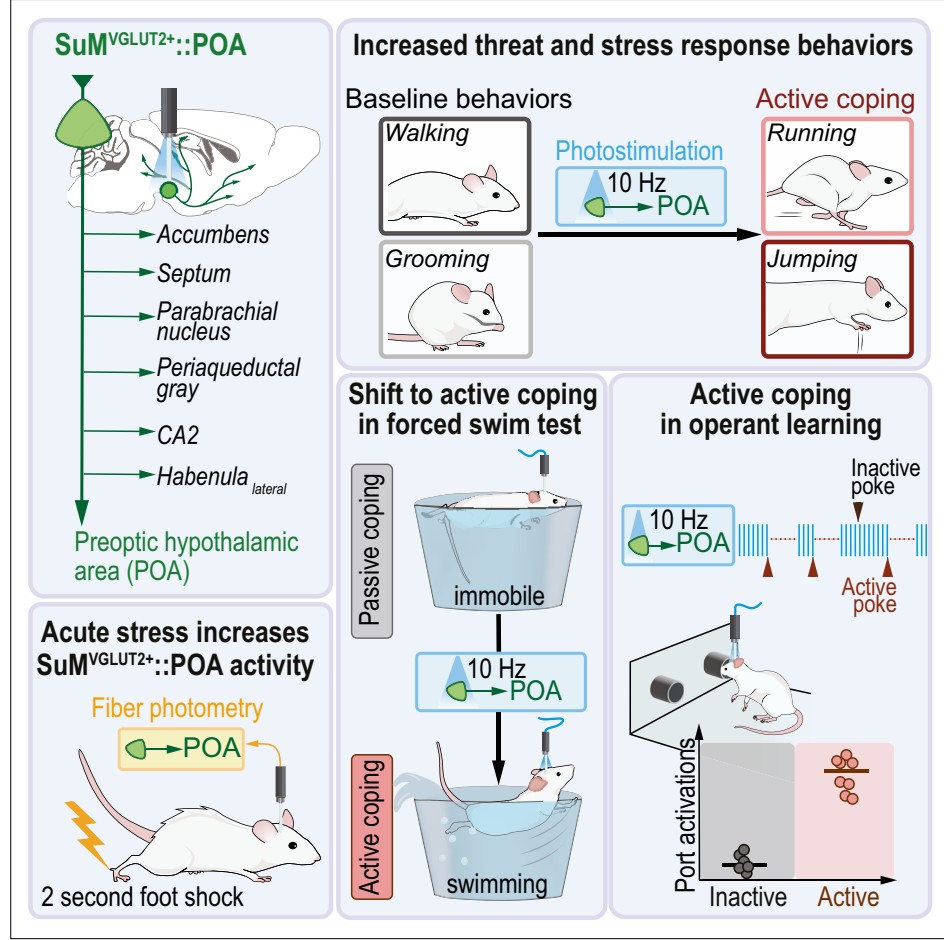

**Figure 9.** Graphical summary. The presented studies show SuM^VGLUT2+^::POA neurons and a subset of neurons in SuM with broad arborizations to multiple brain regions involved in responding to threat and are recruited by multiple threating stressors including foot shock. Activation of SuM^VGLUT2+^::POA neurons is sufficient to evoke active coping-like behaviors in the absence of threating stressors, convert passive coping strategies to active coping strategies, drive aversion, and evoke performance of an operant task under a negative reinforcement paradigm.

neurons increased active coping during ongoing stress in a forced swim assay (*Figure 7H*). Finally, examining the conflicting behaviors required for threat response and feeding, we found SuM$^{VGLUT2+}$::POA neurons were suppressed during feeding and that activation of SuM$^{VGLUT2+}$::POA neurons blocked feeding activity (*Figure 8*). Thus, SuM$^{VGLUT2+}$::POA neurons contribute to affective component of stress, promote active coping behaviors, and shift behavior to shift threat responsivity away from feeding.

The presented results demonstrate SuM$^{VGLUT2+}$::POA neurons act as a distinct subset of neurons projecting to many but not all areas receiving projections from the total SuM$^{VGLUT2+}$ population. Notably, mHb and DG projections were present in the SuM$^{VGLUT2+}$ population but absent in the SuM$^{VGLUT2+}$::POA projections (*Figure 3K*), consistent with a separation of DG and CA2 projecting SuM neurons (*Chen et al., 2020*). The extensive arborization of SuM$^{VGLUT2+}$::POA neurons challenges the often-employed neural circuit concept that projection from a single area to a another area mediates or modifies a behavior. The projections to PVT, Hb, and PAG are of particular interest because these areas regulate stress coping (*Andalman et al., 2019*; *Berton et al., 2007*; *Tovote et al., 2016*; *Keay and Bandler, 2001*). Dopamine receptor 2 expressing neurons in the PVT have been implicated in promoting active coping behaviors and may be a potential target of SuM$^{VGLUT2+}$::POA neurons (*Penzo et al., 2015*; *Ma et al., 2021*). The habenula also regulates active vs passive coping behaviors, and the lHb functions in aversive processing, making the selective projections of SuM$^{VGLUT2+}$::POA neurons to the lHb of similar interest (*Andalman et al., 2019*; *Lee et al., 2010b*; *Li et al., 2011*; *Lazaridis et al., 2019*; *Mondoloni et al., 2022*). In addition, the PAG, which receives projections from SuM$^{VGLUT2+}$::POA neurons, has been implicated in regulating defensive behaviors and coping strategies (*Berton et al., 2007*; *Tovote et al., 2016*). The arborization to these and other areas position SuM$^{VGLUT2+}$::POA neurons to be a central hub for regulating behavioral responses to threatening stressors by concurrent recruitment of multiple areas. Understanding the roles of individual projections in contributing to the overall behavioral response will require further studies.

Superficially, the aversiveness of SuM$^{VGLUT2+}$::POA neuron stimulation we report contrasts with a recent study that found photostimulation of SuM neuron projections could be reinforcing. Photostimulation of projections from SuM to the septum drove place preference (septum), but the same report found photostimulation of SuM projections in the PVT was aversive (*Kesner et al., 2021*). We report here that SuM$^{VGLUT2+}$::POA neurons also project to the PVT and the septum (*Figure 2*). A plausible interpretation is that SuM contains separable, molecularly defined populations of neurons which gate a host of distinct behavioral strategies through their connectivity and/or co-transmitter content. Delineating the overarching circuitry will require further in depth studies, but our results together with recent studies establish the SuM as an important and poorly understood node which regulates both appetitive and aversive motivated behavior.

We addressed the challenge of potential confounds from locomotor activity inherent in active coping behaviors, which is of special concern here because a population of SuM neurons have been found to correlate with velocity of movement. To do so, we used multiple behavioral assays together with fiber photometry and photostimulation. We found multiple threatening stressors (dunk, shock, ambush) with differing evoked behaviors all recruit SuM$^{VGLUT2+}$::POA neurons (*Figure 3*). We also found increased recruitment of SuM$^{VGLUT2+}$::POA neurons during active coping behaviors (*Figures 3 and 7*). We did not find evidence for recruitment of SuM$^{VGLUT2+}$::POA neurons during elevated locomotion in two cohorts in assays of free movement (*Figure 3K–M* and *Figure 3—figure supplement 2*). These data are in contrast to recent studies that found activity of some neurons in SuM correlated future velocity (*Farrell et al., 2021*). We carefully examined the behaviors elicited by activation of SuM$^{VGLUT2+}$::POA neurons both in absence of stressors (*Figure 4*) and during ongoing stressors (*Figure 7*) and found that stimulation of SuM$^{VGLUT2+}$::POA neurons promoted active coping behaviors, which involved increased physical activity. We examined if SuM$^{VGLUT2+}$::POA neurons could also drive performance of an operant task which would be impaired if activation of SuM$^{VGLUT2+}$::POA neurons lead to fixed escape behaviors. We found that the mice effectively performed the operant task, extensively activating the active port, instead of the inactive port. Taken together, the results from the behavioral tests indicate that SuM$^{VGLUT2+}$::POA promote flexible stressor appropriate coping behaviors and not generalized locomotion.

Although neurons in SuM can release both GABA and glutamate in the dentate gyrus, our anatomic studies indicate that SuM$^{VGLUT2+}$::POA neurons were discrete from GABAergic neurons and do not

project the dentate gyrus (*Figure 2* and *Figure 1—figure supplement 2*; *Billwiller et al., 2020*; *Hashimotodani et al., 2018*). Thus, SuM$^{VGAT}$ neurons could be a functionally distinct population. We tested the effects of photostimulation of SuM$^{VGAT}$ neuron and, in contrast to SuM$^{VGLUT2+}$::POA neurons, found no effect on place aversion, minor decreases in locomotion, and no significant change in active coping behaviors during stress (FST) during stimulation. Surprisingly, immediately after photostimulation stopped, active coping behavior increased dramatically (*Figure 7I*). These data demonstrate functional separation of SuM$^{VGAT}$ and suggest connection between the SuM$^{VGLUT2+}$::POA and SUM$^{VGAT+}$ populations that require further investigation to elucidate.

In conclusion, SuM$^{VGLUT2+}$::POA neurons arborize to multiple areas involved in stress and threat response that promote active coping behaviors. Passive coping strategies have been associated with unescapable stress, anhedonia, and depression (*Southwick et al., 2005*; *Cryan, 2005*; *Knoll and Carlezon, 2010*; *Ebner and Singewald, 2017*). Targeting neuromodulation of a circuit able to act across many brain areas may represent a therapeutic avenue for common psychiatric conditions, and SuM$^{VGLUT2+}$::POA neurons are a newly identified node in critical approach-avoidance circuitry.

# Materials and methods

## Key resources table

| Reagent type (species) or resource | Designation | Source or reference | Identifiers | Additional information |
|---|---|---|---|---|
| Strain, strain background (*Slc17a6*$^{tm2(cre)Lowl}$/J, C57BL/6;FVB;129S6) | *Slc17a6*$^{tm2(cre)Lowl}$/J (VGLUT2-Cre) | The Jackson Laboratory | RRID:IMSR_JAX:016963 | Males and females |
| Strain, strain background (B6;129S-*Slc17a6*$^{tm1.1(flpo)Hze}$/J, C57BL/6 J) | B6;129S-*Slc17a6*$^{tm1.1(flpo)Hze}$/J (VGLUT2-Flp) | The Jackson Laboratory | RRID: IMSR_JAX: 030212 | Males and females |
| Strain, strain background (B6.Cg-*Gt(ROSA)26Sor*$^{tm14(CAG-tdTomato)Hze}$/J, C57BL/6 J) | B6.Cg-*Gt(ROSA)26Sor*$^{tm14(CAG-tdTomato)Hze}$/J (Ai14) | The Jackson Laboratory | RRID: IMSR_JAX: 007914 | Males and females |
| Strain, strain background (B6J.129S6(FVB)-*Slc32a1*$^{tm2(cre)Lowl}$/MwarJ, C57BL/6;FVB;129S6) | B6J.129S6(FVB)-*Slc32a1*$^{tm2(cre)Lowl}$/MwarJ (VGAT-Cre) | The Jackson Laboratory | RRID: IMSR_JAX: 028862 | Males and females |
| Strain, strain background (C57BL/6 J, C57BL/6 J) | C57BL/6 J | The Jackson Laboratory | RRID: IMSR_JAX: 000664 | Males and females |
| Antibody | Alexa Fluor 488 Goat anti-Rabbit IgG (Goat Polyclonal) | Invitrogen | Cat# A11008 | IHC (1:2500) |
| Antibody | Phospho-c-Fos (Ser32) (D82C12) XP (Rabbit Monoclonal) | Cell Signaling Technology | Cat# 5348 | IHC (1:500) |
| Recombinant DNA reagent | AAV5-EF1a-DIO-hChR2(H134R)-EYFP (2.5×10$^{13}$ vg/ml) | Washington University Hope Center Viral Vector Core | N/A | |
| Recombinant DNA reagent | AAV2-Retro-DIO-ChR2-eYFP (2.8×10$^{12}$ vg/ml) | Washington University Hope Center Viral Vector Core | N/A | |
| Recombinant DNA reagent | AAV2-retro-FLEX-tdTomato (7×10$^{12}$ vg/ml) | Addgene | RRID: Addgene_28306-AAVrg | |
| Recombinant DNA reagent | AAV5-DIO-ChR2-eYFP (1.4×10$^{13}$ vg/ml) | Washington University Hope Center Viral Vector Core | N/A | |
| Recombinant DNA reagent | AAV2-Retro-EF1a-fDIO-cre (7×10$^{12}$ vg/ml) | Addgene | RRID: Addgene_121675-AAVrg | |
| Recombinant DNA reagent | AAV5-EF1a-Nuc-flox(mCherry)-EGFP (5.7×10$^{12}$ vg/ml) | Addgene | RRID: Addgene_112677-AAV5 | |
| Recombinant DNA reagent | Retro-AAV2-EF1a-Nuc-flox(mCherry)-EGFP (7×10$^{12}$ vg/mL) | Addgene | RRID: Addgene_112677-AAVrg | |
| Recombinant DNA reagent | AAV2-retro-EF1a-DIO-eYFP (3×10$^{13}$ vg/ml) | Washington University Hope Center Viral Vector Core | N/A | |
| Recombinant DNA reagent | AAV-retro-EF1a-Flpo (1.02×10$^{13}$ GC/mL or 7×10$^{12}$ vg/ml) | Addgene | RRID: Addgene_55637-AAVrg | |
| Recombinant DNA reagent | AAV9-EF1a-fDIO-cre (2.5×10$^{13}$ GC/mL or 1×10$^{13}$ vg/ml) | Addgene | RRID: Addgene_121675-AAV9 | |

*Continued on next page*

*Continued*

| Reagent type (species) or resource | Designation | Source or reference | Identifiers | Additional information |
|---|---|---|---|---|
| Recombinant DNA reagent | Retro-AAV2-EF1a-Cre (2.1×10^13 GC/mL) | Addgene | RRID: Addgene_55636-AAVrg | |
| Recombinant DNA reagent | AAV9-syn-FLEX-GCaMP7s-WPRE (1.2×10$^{13}$ GC/ml) | Addgene | RRID: Addgene_104487-AAV9 | |
| Recombinant DNA reagent | AAV5-EF1a-DIO-eYFP (1.8x10$^{13}$ vg/ml) | Washington University Hope Center Viral Vector Core | N/A | |
| Recombinant DNA reagent | AAV2 retro-Ef1a-DIO-mCherry 4.0×10$^{12}$ vg/ml | University of Carolina Vector Core | N/A | |
| Chemical compound, drug | 4% paraformaldehyde | J.T. Baker, Avantor | S898-09 | |
| Chemical compound, drug | Goat Serum | Sigma Aldrich | G9023 | |
| Software, algorithm | Bonsai | https://bonsai-rx.org/ | N/A | |
| Software, algorithm | Ethovision | Noldus | N/A | |
| Software, algorithm | GraphPad Prism | GraphPad Software | N/A | |
| Software, algorithm | Labgym | https://github.com/umyelab/LabGym; *Hu et al., 2023*; *Satpathy et al., 2024* | N/A | |
| Software, algorithm | DeepLabCut | https://github.com/DeepLabCut/DeepLabCut; *Mathis et al., 2018*; *Nath et al., 2019*; *DeepLabCut, 2024* | N/A | Version 2.2 |
| Other | Vectashield Hardset Antifade Mounting Medium with DAPI | Vector Laboratories | Cat# NC9029229 | Mounting medium with DAPI for fluorescence microscopy. |

For further information regarding data, reagents and resources, contact Aaron Norris, norrisa@wustl.edu.

## Experimental model and subject details

Adult (25–35 g, at least 8 weeks of age upon experimental use) male and female VGLUT-Cre (RRID: IMSR_JAX:016963), VGLUT2-Flp (RRID: IMSR_JAX: 030212), Ai14 (RRID: IMSR_JAX: 007908), VGAT-Cre (RRID: IMSR_ JAX: 028862) and C57BL/6 J (RRID: IMSR_JAX: 000664) mice (species *Mus musculus*) were group housed (no more than five littermates per cage) in a 12 hr:12 hr light:dark cycle room with food and water ad libitum (*Vong et al., 2011*). Cre +and Cre- littermates were used in the experiments unless otherwise noted. The Washington University Animal Care and Use Committee approved all procedures which adhered to NIH guidelines.

All data available in manuscript and supporting files.

## Stereotaxic surgery

Injections and implantations were done as described previously (*Norris et al., 2021*). Briefly, in an induction chamber, mice were anesthetized (4% isoflurane) before being placed in a stereotaxic frame (Kopf Instruments). Anesthesia was maintained with 2% isoflurane. Mice were then injected unilaterally or bilaterally, depending on the combination of virus(es) used and brain regions injected. A blunt needle Neuros Syringe (65457–01, Hamilton Con.) and syringe pump (World Precision Instruments) were used to perform the injection schemes below. After surgery, a warmed recovery chamber housed the animal while it recovered from anesthesia before being returned to its home cage.

| Brain Region /Coordinates | Virus Volume | Virus |
|---|---|---|
| SuM<br>(AP –2.7–2.85, ML +0.05, DV –4.3–4.5) | 50–300 nl | AAV5-EF1a-DIO-hChR2(H134R)-EYFP<br>AAV2-retro-DIO-ChR2-eYFP<br>AAV5-EF1a-Nuc-flox(mCherry)-EGFP<br>AAV9-EF1a-fDIO-cre<br>AAV9-syn-FLEX-GCaMP7s-WPRE<br>AAV5-EF1a-DIO-eYFP |
| POA, bilateral<br>(AP +0.45, ML ±0.5, DV –5.35) | 50–150 nl | AAV2-retro-EF1a-fDIO-cre<br>Retro-AAV2-EF1a-Nuc-flox(mCherry)-EGFP<br>AAV2-retro-EF1a-DIO-eYFP<br>AAV2-retro-EF1a-Flpo<br>AAV2 retro-Ef1a-DIO-mCherry |
| Septum<br>(AP +0.9, ML +0.3, DV –3.5) | 100 nl | AAV2-retro-FLEX-tdTomato |
| LPAG, unilateral<br>(AP –4.65, ML +0.6, DV –2.85) | 100 nl | AAV2-retro-FLEX-tdTomato |
| Acb, unilateral,<br>(AP +1.4,+0.6 ML, DV –4.75) | 100 nl | AAV2-retro-EF1a-DIO-eYFP |
| PVT, unilateral,<br>(AP –1.5, ML 0.0, DV –2.85) | 100 nl | AAV2-retro-EF1a-DIO-eYFP |

Injections were made at a rate of 50 nl/min, with the injection needle being withdrawn 5 min after the end of the infusion. Fiber optics for photostimulation or optical fibers for fiber photometry were implanted after injections for all behavioral experiments.

Fiber optic implants for photostimulation were fabricated as previously described using 200 µm glass fibers and implanted midline over SuM (*Norris et al., 2021*; *Sparta et al., 2011*). For implantation, the skull was cleaned and etched with OptiBond (Kerr) and the fiber was cemented to the skull with Tetric N-Flow (Ivoclar Vivadent). Blue light was used to cure and harden cement. Mice were allowed to recover for at least 7 days before the start of behavioral experiments. The same process was used for implantation for fiber photometry fibers, which were purchased from Neurophotometrics and trimmed to length.

## Anatomical tracing

For anterograde viral tracing experiments, virus was injected at least 6 weeks prior to transcardial perfusions with 4% paraformaldehyde to allow for anterograde transport of the fluorophore. AAV5-EF1a-DIO-hChR2eYFP or AAV5-EF1a-DIO-eYFP were used. Alternatively, to label only SuM[V-GLUT2+]::POA neurons for anterograde tracing, Retro-AAV2-DIO-eYFP or Retro-DIO-ChR21eYFP was injected into the POA with AAV-fDIO-Cre injected in the SuM. A minimum of 6 weeks was allowed prior to sacrifice, harvesting or brains, and sectioning (30 µM). Serial 30 µM sections approximately 60 µM apart were examined. For retrograde studies, viruses were injected (see figure legends and text for specific viruses) at the targeted site and table for specific viruses (*Fenno et al., 2014*; *Bäck et al., 2019*; *Lee et al., 2010a*). Three weeks were allowed to elapse prior to harvesting brains following injections. Images were obtained on a Leica DM6 B upright microscope and processed using Thunder imaging station (Leica).

## Brain clearing and light sheet microscopy

Tissue clearing and imaging was carried out on brains collected and fixed in 4% PFA as described above by LifeCanvas Technologies. Briefly, brains were fixed using SHIELD post ix and cleared for 7 days in SmartClear II Pro. Index matched with EASYIndex at room temperature. Samples were mounted ventral side up and imaged at 3.6 x with pixel size 1.8x1.8 mm, axial resolution <4 µm, z step 4 µm in 488 nm channel. Fos was labeled by Alexa Flour 488. SuM boundaries were defined by –2.6 to –2.95 rostral to Bregma. The medial mammillary nucleus and the mammillary recess of the 3rd ventricle marked the medial and ventral boundaries, while fornix marked the lateral, and fasciculus retroflexus marked the dorsal boundaries. Images were quantified by a trained laboratory member who was blind to the experimental conditions.

## Immunohistochemistry

Mice were intracardially perfused with 4% PFA and then brains were sectioned (30 microns) and placed in 1 x PB until immunostaining. Free-floating sections were washed three times in 1 x PBS for

10 min intervals. Sections were then placed in blocking buffer (0.5% Triton X-100% and 5% natural goat serum in 1z PBS) for 1 hr at room temperature. After blocking buffer, sections were placed in primary antibody rabbit Phospho-c-Fos (Ser32) antibody (1:500, Cell Signaling Technology) overnight 4 °C temperature. After 3x10 min 1 x PBS washes, sections were incubated in secondary antibody goat anti-rabbit Alexa Fluor 488 (1:2500, Invitrogen) for 2 hr at room temperature. Sections were washed in 1 x PBS (3x10 min) followed by 2x10 min 1 x PB washes. After immunostaining, sections were mounted on Super Frost Plus slides (Fisher) and covered with Vectashield Hard set mounting medium with DAPI (RRID:AB_2336788, Vector Laboratories) and cover glass prior to being imaged.

## Imaging and cell quantification

'The Mouse Brain in Stereotaxic Coordinates' provided the framework to label brain sections relative to bregma (*Paxinos, 2019*). A Leica DM6 B epifluorescent microscope was used to image all sections. For eYFP visualization, a YFP filter cube (Excitation: 490–510, Dichroic: 515, Emission: 520–550) was used and for tdTomato visualization, a Texas Red Filter Cube (Excitation: BP 560/40, Dichroic: LP 585, Emission: BP 630/75) was used. Fos was labeled by Alexa Flour 488. SuM boundaries were defined by –2.6 to –2.95 rostral to Bregma. The medial mammillary nucleus and the mammillary recess of the 3rd ventricle marked the medial and ventral boundaries while fornix marked the lateral, and fasciculus retroflexus the dorsal boundaries. Images were quantified by a trained laboratory member who was blind to the experimental conditions.

## Forced-swim test for cFos examination

For stress induction via forced swim (*Figure 3—figure supplement 1*; *Commons et al., 2017*; *Porsolt et al., 1979*), mice were individually placed in a cylindrical container (18 cm in diameter) filled with water at 25+/-1 °C for 15 min. Prior to stress force swimming, mice were habituated to the arenas for 3 days prior to the beginning of FST to minimize stress. Ninety min after forced swim, mice were injected with ketamine and xylazine and were intracardially perfused. Control mice were brought to the behavioral testing area but remained in the home cage until perfusion. Water was replaced after every animal. To examine SuM-POA glutamatergic projections, VGlut2-Cre+littermates were injected with AAV2 retro-Ef1a-DIO-mCherry into the POA.

## Fiber photometry

For all fiber photometry experiments, the same strategy to selectively label SuM^VGLUT2+^::POA neurons was used. VGLUT2-Flp mice were injected bilaterally in POA with AAV-Retro-EF1a-fDIO-Cre and with AAV-syn-FLEX-GCaMP7s-WPRE in SuM. After 2 weeks, mice were implanted with fiber-optic cannulas (200 μm) in SuM (D/V –4.3–5; *Dana et al., 2019*). Mice recovered a minimum of 1 week prior to behavioral testing. Recording of Ca$^{2+}$-dependent and isosbestic signals were obtained using previously described methods with Bonsai software and FP3002 (Neurophotometrics; *Parker et al., 2019*; *Martianova et al., 2019*). 470 and 415 nm LEDs were used to record interleaved isosbestic and Ca$^{2+}$-dependent signals following the manufacturer directions.

For repeated forced swim experiments (dunk tank), mice were placed on top of a square wire mesh platform inside a custom-made plastic rectangular enclosure (20 cm x 20 cm x 23 cm) filled with water 30+/-1 °C. The square wire mesh platform could be raised and lowered without touching the animal. Mice were habituated for 2 days before the test day. Each day, mice were tethered to the fiber optic patch cable and placed on top of the platform for 30 min. On test day, mice were tethered and placed on top of the platform for 30 min (habituation) before the beginning of the 1st trial. After 30 min, for each trial the platform was lowered for 30 s and raised after 30 s with a rest of 2 mins between trials for a total of 10 trials ('dunks'). After 10 trials, an additional 2 min were recorded before removing the animal from the enclosure, patted the animals with paper towels, and placed back in its home cage. During testing, mice were tethered to a fiberoptic patch cable that was attached to a counterbalanced arm that prevented downward force on the animal. Water was replaced after each animal. Time mobile was quantified by a trained observer for the ten 30 s trials.

For foot shock stress testing, mice were individually placed inside a custom-made clear plastic box (15.24 L x 13.34 W x 14 H cm) inside a sound-attenuated cabinet. A speaker was placed 4 cm above the chamber for the delivery of auditory cues (75 dB). A constant current aversive stimulator (ENV-414S) delivered foot shocks through a grid floor (0.7mA). Five shocks of 2 s were delivered after a

30 s tone. Intertrial interval ranged from 90 to 180 s. After the session, the animal was removed from the chamber and placed back in its home cage. The chamber and grid were wiped down with 70% ethanol between animals.

For open field testing velocity measurements, mice were tethered to a fiberoptic patch cable and placed inside a custom-built square arena (50.8 cm x 50.8 cm x 50.8 cm). Mice were allowed to explore the arena for a 20-min session. Velocity was quantified using scripts in Bonsai 2.4. Bedding in the arena was replaced between animals, and the floors and walls of the arena were wiped down with 70% ethanol.

Similar to the repeated forced swim experiments, forced swim mice were placed on top of a square wire mesh platform that could be raised and lowered inside a custom-made plastic rectangular enclosure (20 cm x 20 cm x 23 cm) filled with water at 25+/-1 °C. The platform was lowered at the beginning of the test for 15 min and raised at the conclusion of the test. During testing, mice were tethered to a fiberoptic patch cable that was attached to a counter balanced arm that prevented downward force on the animal. Water was replaced between animals and the enclosure was wiped down with 70% ethanol and rinsed with water. Time immobile, hindpaw swim, swimming, and climbing were quantified using LabGym (*Hu et al., 2022*).

To measure the activity of SuM$^{VGLUT2+}$::POA neurons in response to a stressor that mice may encounter in their natural habitat (*Nyffeler, 2018*), a remote-controlled spider was used to simulate an ambush of a potential predator. The remote-controlled spider (17 cm x 16 cm; Amazon) was placed inside a polylactic acid (PLA) enclosure (19 cm x 20 cm x 23 cm) that was then placed inside a custom-built square arena (50.8 cm x 50.8 cm x 50.8 cm). Mice were tethered to a fiberoptic patch cable and placed inside the arena for 10 min. Baseline activity was recorded for 5 min before the ambush. After 5 min, the animal was ambushed once it moved in close proximity to the spider's enclosure (*Video 4*). Velocity was measured and analyzed using DeepLabCut within Bonsai (*Mathis et al., 2018*). The arena and spider's enclosure were wiped down with 70% ethanol between animals.

For experiments that examined recruitment of SuM$^{VGLUT2+}$::POA neurons during consummatory behavior, mice were food deprived or given ad lib access to food for 24 hr prior to testing. On test day, mice were placed inside an 18 cm diameter round arena and allowed to habituate for 30 min before they were tethered to the fiberoptic path cable. Once tethered, mice were placed inside the arena for 20 min. Baseline activity was recorded for 5 min before the introduction of the chow pellet and for 15 min after the chow pellet was introduced. Chow pellets were weighed before and after the 20-min trial. The difference is reported as food eaten. The same procedure was followed for control mice with the only difference being that these animals were not food deprived for 24 hr.

For fiber photometry data analysis, the interleaved isosbestic and Ca$^{2+}$-dependent signals were recorded at 60 fps (30 fps each). Deinterleaved signals were analyzed using methods as previously reported (*Martianova et al., 2019*). Briefly, raw signals were smoothed using a moving average, fitted with an exponential curve using a non-linear least squares function for baseline correction, signals were standardized using the mean value and standard deviation (Z-Score), the standardized isosbestic and Ca2 +signals were scaled a non-negative robust linear regression, and normalized dF/F was calculated. In experiments shown in *Figures 7A and 8* E-F there was sustained step drop evident in the Ca$^{2+}$-dependent signal reflecting change in population activity because it was not seen in the isosbestic signal. The nature and duration of the change of the signal precluded fitting a curve to the Ca$^{2+}$-dependent signal. In these cases, we show both the Ca$^{2+}$-dependent and isosbestic signals. Z-scores were calculated without baseline correction for both Ca$^{2+}$-dependent and isosbestic signals based on the variability in the baseline state.

## Real-time place aversion testing

For real-time place preference testing with optogenetic photostimulation, we used custom-made, unbiased, balanced two-compartment conditioning apparatus (52.5x25.5 x 25.5 cm) as described previously (*Norris et al., 2021*; *Stamatakis and Stuber, 2012*). Mice were tethered to a patch cable that allowed free access to the entire arena for 30 min. Entry into one compartment triggered photostimulation, 1 Hz, 5 Hz, 10 Hz, or 20 Hz (473 nm laser, 10ms pulse width), that persisted while the mouse remained in the light paired side. The side paired with photostimulation was counterbalanced across mice. Ordering was counterbalanced with respect to stimulation frequency and placement. Bedding in the behavior apparatus was replaced between every trial, and the floors and walls of the

apparatus were wiped down with 70% ethanol. Time spent in each chamber and total distance traveled for the entire 30-min trial was measured using Ethovision 10 (Noldus Information Technologies). For optogenetic stimulation of VGAT neurons during RTTA, Cre + littermates were injected either with AAV5-EF1a-DIO-hChR2(H134R)-EYFP or with AAV5-EF1a-DIO-eYFP as control, and stimulation was provided at 10 Hz.

### Light /dark choice

For light/dark choice, the same arenas as used for real-time preference testing were modified and used as previously described (*Luskin et al., 2021*). On the light side, a small sport light was placed overhead, and the walls were covered with white laminated paper. Light levels on this side measure 580–590 lux. For the dark side, an infrared spotlight was placed over head, to allow for video tracking of the mice. The walls were covered with matte black plastic. Light levels in the center of the dark side measure 100–110 lux. Animals were recorded using a USB web cam without an infrared filter. For real-time aversion testing, photostimulation was paired with the dark side of the arena as described above. For anxiety-like behavior testing, stimulation was provided uniformly during the trials. Time spent in each chamber and total distance traveled for the entire 30-min trial was measured using Ethovision 10.

### Observational behavioral assay

To observe behaviors evoked by photostimulation of SuM$^{VGLUT2+}$::POA neurons, mice were habituated for at least 3 days prior to testing to a round (18 cm diameter), clear arena with counterbalanced optical commutators to minimize the impact of the head tether on movement. Testing occurred after habitation, and approximately 2 cm of bedding material was placed in the arena. Behavior was recorded from the side and scored by a blinded observer.

### Negative reinforcer two nose port operant behavior testing

For operant behavior testing, we used Med Associates mouse operant conditioning chamber with dual nose ports and house light as previously described (*Parker et al., 2019*). Briefly, mice were tethered via cantilevered counterweighted optical commutator to a laser light source. A 5-day protocol call was used. Day 1 through four were 10 min trials. Photostimulation was provided at 10 Hz and activation of the active port resulted in a 10 s pause in the photostimulation and activation of the house light inside the arena. On the first four days, each activation of the active port outside of a 10 s pause resulted in a new pause. On day 5 was a 30-min trial using a progressive ratio protocol the number activation of the active port to generate a pause increased with each activation based on the $number\ of\ activations\ (j) = \left[5e^{(0.2j)} - 5\right]$ round to the nearest integer generating the schedule 1, 2, 4, 6, 9, 12, 15, 20, 25, 32, 40, 50….(*Richardson and Roberts, 1996*). The trial was limited to 30 min due to concern for animal welfare due to the head tether and confined space. Photostimulation terminated at the conclusion of the trial.

### Forced swim with photostimulation

Forced swim trials with optogenetic stimulation were done as a two-day test. On the first day, all mice were subjected to a 15 min forced swim, dried, and returned to the home cage. On the second day, they were tethered and subjected to a 6 min forced swim, divided into three periods, each two mins in length: pre stimulation, stimulation, and post stimulation. Ten Hz photostimulation was provided during the trial. Mice were closely monitored during each trial of swimming. Trials were recorded and scored by a blinded observer for time spent immobile on the second day of the test. For forced swim trials with optogenetic stimulation of VGAT neurons, Cre +littermates were injected either with AAV5-EF1a-DIO-hChR2(H134R)-EYFP or with AAV5-EF1a-DIO-eYFP as control.

### Feeding after food deprivation

For tests involving brief access to food after deprivation, the same 18 cm diameter round arenas with counterbalanced optical commutators were used. Mice were habituated to the arenas for a minimum of three days prior to testing. Mice were food deprived by removal of food from the home cage 24 hr prior to testing. Mice were placed in the arena and allowed to habituate prior to introduction of a chow pellet. The 20-min trial with the foot pellet was recorded and scored by a blinded observer. Chow pellets were weighed before and after the 20-min trial. The difference is reported as food eaten.

## Open field test

For Open Field testing, we used a purpose-built 20 in square behavior arena. Mice were tethered to a patch cable and placed into the behavioral arena. The laser frequency was set to 10 Hz and was left on for 20 min. Distance moved for the 20-min trial was quantified using Ethovision 10. Bedding in the arena was replaced between every trial, and the floors and walls of the arena were wiped down with 70% ethanol. Similarly, for optogenetic stimulation of VGAT neurons, Cre +littermates were injected either with AAV5-EF1a-DIO-hChR2(H134R)-EYFP or with AAV5-EF1a-DIO-eYFP as control.

## Statistical analyses

Statistical analyses were conducted using GraphPad Prism software. Data are shown as mean ± SEM, except for Z-scores which are shown as ±95% confidence interval, as noted in text. Values for individual p values are given in the text and figure legends. Significance was held at α less than 0.05. In cases of multiple comparisons, Bonferroni method was used to correct for multiple comparisons. Paired testing was used when comparing within cohorts with repeated measurements and unpaired between cohorts. All 'n' values represent the number of animals in a particular group for an experiment. For fiber photometry statistical analysis, the mean signal of the baseline and stimulus windows was used, and comparisons were made using the Wilcoxon ranked-sum test, with $\alpha=0.95$. To analyze the change in the $Ca^{2+}$-dependent signal, a t-confidence interval method was used, in which 95%, 99%, and 99.9% confidence intervals were calculated for windows preceding and subsequently following the described point of interest (*Jean-Richard-Dit-Bressel et al., 2020*). Differences were considered significant if the null hypothesis (zero) was not included in the CI. Because exact p values are not calculated using this method, the highest confidence level at which the difference is significant (95%, 99%, or 99.9% CI) is reported instead. Please see *Supplementary file 1* for a table with all statistical analyses conducted.

## Acknowledgements

This work was supported by the NIH through 5R01MH112355 to MRB and P30DA048736 (MRB) and 5K08MH119538 to AJN. Support was also provided by the Hope Center Viral Vectors Core and the Genome Technology Resource Center at Washington University in St. Louis (NIH P30CA91842 and UL1TR000448). This work was supported by a Pilot Project Award from the Hope Center for Neurological Disorders and by the Hope Center Viral Vectors Core at Washington University School of Medicine. The graphic summary illustration (*Figure 9*) was created by Suelynn Ren in association with InPrint at Washington University School of Medicine.

## Additional information

### Funding

| Funder | Grant reference number | Author |
| --- | --- | --- |
| National Institute of Mental Health | 5K08MH119538 | Aaron J Norris |
| National Institute of Mental Health | 5R01MH112355 | Michael R Bruchas |
| National Institute on Drug Abuse | P30DA048736 | Michael R Bruchas |
| Hope Center for Neurological Disorders | | Aaron J Norris |

The funders had no role in study design, data collection and interpretation, or the decision to submit the work for publication.

### Author contributions

Abraham Escobedo, Conceptualization, Formal analysis, Investigation, Writing - original draft, Writing - review and editing; Salli-Ann Holloway, Megan Votoupal, Aaron L Cone, Conceptualization, Data curation, Formal analysis, Investigation; Hannah Skelton, Conceptualization, Formal analysis,

Investigation, Writing - review and editing; Alex A Legaria, Formal analysis, Methodology; Imeh Ndiokho, Data curation, Investigation; Tasheia Floyd, Investigation; Alexxai V Kravitz, Conceptualization, Supervision, Methodology, Writing - review and editing; Michael R Bruchas, Conceptualization, Supervision, Funding acquisition, Writing - review and editing; Aaron J Norris, Conceptualization, Data curation, Formal analysis, Supervision, Funding acquisition, Investigation, Writing - original draft, Writing - review and editing

### Author ORCIDs

Abraham Escobedo http://orcid.org/0009-0004-4418-753X
Salli-Ann Holloway http://orcid.org/0009-0001-7778-5462
Aaron L Cone http://orcid.org/0000-0003-4411-6673
Hannah Skelton http://orcid.org/0009-0009-7322-2900
Alexxai V Kravitz http://orcid.org/0000-0001-5983-0218
Michael R Bruchas http://orcid.org/0000-0003-4713-7816
Aaron J Norris http://orcid.org/0000-0001-7825-1756

### Ethics

The Washington University Animal Care and Use Committee (IACUC) approved all procedures which adhered to NIH guidelines. Animal Welfare Assurance #D16-00245. All surgeries were performed under anesthesia.

Joint Public Review: https://doi.org/10.7554/eLife.90972.3.sa1
Author response https://doi.org/10.7554/eLife.90972.3.sa2

## Additional files

### Supplementary files

• Supplementary file 1. Statistical analyses and values. Statistical analyses conducted for all figures including the type of analysis(es), p values, statistic (i.e. *F*, *t* statistic), and/or multiple comparisons.
• MDAR checklist

### Data availability

All data generated or analyzed during this study are included in the manuscript and supporting files.

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
