## [Editor Report · eLife assessment]

This **important** manuscript investigates the role of a subpopulation of glutamatergic neurons in the suprammamillary nucleus that projects to the pre-optic hypothalamus area in active coping but not locomotor activity. They provide **solid** evidence from experiments using fibre photometry or photostimulation during threatening tasks that these neurons allow animals to produce flexible behaviours in response to stress. This work will be of interest to behavioural and systems neuroscientists.

---

## [Referee Report · Joint Public Review]

Summary:

This important manuscript investigates a subpopulation of glutamatergic neurons in the suprammamillary nucleus that projects to the pre-optic hypothalamus area (SuM-VGLUT2+::POA). First, they define the neural circuitry of these neurons, which contact many stress/threat-associated brain regions. Then they employ fibre photometry to measure the activity of these neurons during various threatening tasks and find the responses correlate well with threat stimuli. Finally, they stimulate these neurons and find multiple lines of evidence that mice find this aversive and will act to avoid receiving this stimulation. In sum, they provide solid evidence that this neuronal population represents a new node in stress response circuitry that allows the animal to produce flexible behaviours in response to stress, which will be of interest to neuroscientists across several sub-fields.

Strengths:

Overall this is a solid manuscript tackling an important question. Coping with stress by an animal in danger is essential for survival. This manuscript identifies a novel population of neurons in the murine supramamillary nucleus (SuM) projecting to the pre-optic hypothalamus area among other regions that is involved in this important process. The evidence to support the conclusions is solid.

Specific strengths:

• The topic is novel.

• The manuscript follows a logical structure and neatly moves through the central story. Several potential alternate interpretations are well-controlled for.

• The manuscript employs an array of different tasks to provide converging evidence for their conclusions.

• The authors provide excellent evidence of the specificity of the function of this neuronal population, both from anatomical studies and from behavioural studies (e.g. demonstrating that activity of gabaergic neurons in the same region does not correlate with behaviours in the same way).

• The study is well-powered (sample sizes are good) and the effects are convincing.

Weaknesses:

* Not all of the reviewer comments were addressed in the manuscript itself, although this was acknowledged in the author's responses to reviewers. One key example is as follows:

* The authors did not entirely address comments related to rigor but they at least acknowledged it. For example, in multiple places they argue that WT, purchased mice are probably not different in baseline behavior compared to Vgltu2-IRES-Cre because it is unlikely that adding the IRES-Cre will change behavior. However, they do not acknowledge that transgenic lines are not from the exact same genetic background and generation number, and there is ample evidence in the literature that transgenic mice on a B6J background can differ in basal phenotypes from one another and B6J. In one place they show some basal behavior, at least in heat map form though not quantified. Had the authors decided to apply this more pervasively, it would have made the story even more compelling in terms of a stress/threat-induced phenotype.

Comments on revised version from the Reviewing Editor:

The authors have done a thorough job of answering the reviewer queries, and a good job of explaining why they have not answered a particular point. Indeed, there is so much additional information in response to the reviewers that I hope readers of the manuscript will read the reviews and responses as well! I think they add a lot.

---

## [Author Response]

The following is the authors’ response to the original reviews.

We would like to thank the reviewers for their insightful comments and recommendations. We have extensively revised the manuscript in response to the valuable feedback. We believe the results is a more rigorous and thoughtful analysis of the data. Furthermore, our interpretation and discussion of the findings is more focused and highlights the importance of the circuit and its role in the response to stress. Thank you for helping to improve the presented science.

Key changes made in response to the reviewers comments include:

• Revision of statistical analyses for nearly all figures, with the addition of a new table of summary statistics to include F and/or t values alongside p-values.

• Addition of statistical analyses for all fiber photometry data.

• Examination of data for possible sex dependent effects.

• Clarification of breeding strategies and genotype differences, with added details to methods to improve clarity.

• Addressing concerns about the specificity of virus injections and the spread, with additional details added to methods.

• Modification of terminology related to goal-directed behavior based on reviewer feedback, including removal of the term from the manuscript.

• Clarification and additional data on the use of photostimulation and its effects, including efforts to inactivate neurons for further insight, despite technical challenges.

• Correction of grammatical errors throughout the manuscript.

**Reviewer 1:**
Despite the manuscript being generally well-written and easy to follow, there are several grammatical errors throughout that need to be addressed.

Thank you for highlighting this issue. Grammatical errors have been fixed in the revised version of the manuscript.

Only p values are given in the text to support statistical differences. This is not sufficient. F and/or t values should be given as well.

In response to this critique and similar comments from Reviewer 2, we re-evaluated our approach to statistical analyses and extensively revised analyses for nearly all figures. We also added a new table of summary statistics (Supplemental Table 1) containing the type of analysis, statistic, comparison, multiple comparisons, and p value(s). For Figures 4C-E, 5C, 6C-E, 7H-I, and 8H we analyzed these data using two-way repeated measures (RM) ANOVA that examined the main effect of time (either number of sessions or stimulation period) in the same animal and compared that to the main effect of genotype of the animal (Cre+ vs Cre-), and if there was an interaction. For Supplemental Figure 7A we also conducted a two-way RM ANOVA with time as a factor and activity state (number of port activations in active vs inactive nose port) as the other in Cre+ mice. For Figures 5D-E we conducted a two-way mixed model ANOVA that accounted and corrected for missing data. In figures that only compared two groups of data (Figures 5F-L, 6F, 8C-D, 8I, and Supp 6F-G) we used two-tailed t-test for the analysis. If our question and/or hypothesis required us to conduct multiple comparisons between or within treatments, we conducted Bonferroni’s multiple comparisons test for post hoc analysis (we note which groups we compared in Supplemental Table 1). For figures that did or did not show a change in calcium activity (Figure 3G, 3I-K, 7B, 7D-E, 8E-F), we compared waveform confidence intervals (Jean-Richard-Dit-Bressel, Clifford, McNally, 2020). The time windows we used as comparison are noted in Supplemental Table 1, and if the comparisons were significant at 95%, 99%, and 99.9% thresholds.

None of prior comparisons in prior analyses that were significant were found to have fallen below thresh holds for significance. Of those found to be not significantly different, only one change was noted. In Figure 6E there was now a significant baseline difference between Cre+ and Cre- mice with Cre- mice taking longer to first engage the port compared to Cre+ mice (p=0.045). Although the more rigorous approach the statistical analyses did not change our interpretations we feel the enhanced the paper and thank the reviewer for pushing this improvement.

Moreover, the fibre photometry data does not appear to have any statistical analyses reported - only confidence intervals represented in the figures without any mention of whether the null hypothesis that the elevations in activity observed are different from the baseline.This is particularly important where there is ambiguity, such as in Figure 3K, where the spontaneous activity of the animal appears to correlate with a spike in activity but the text mentions that there is no such difference. Without statistics, this is difficult to judge.

Thank you for highlighting this critical point and providing an opportunity to strengthen our manuscript. We added statistical analyses of all fiber photometry data using a recently described approach based on waveform confidence intervals (Jean-Richard-Dit-Bressel, Clifford, McNally, 2020). In the statistical summary (Supplemental Table 1) we note the time window that we used for comparison in each analysis and if the comparisons were significant at 95%, 99%, and 99.9% thresholds. Thank you from highlighting this and helping make the manuscript stronger.

With respect to Figure 3K, we are not certain we understood the spike in activity the reviewer referred to. Figure 3J and K include both velocity data (gold) and Ca2+ dependent signal (blue). We used episodes of velocity that were comparable to the avoidance respond during the ambush test and no significant differences in the Ca2+ signal when gating around changes in velocity in the absence of stressor (Supplemental Table1). This is in contrast to the significant change in Ca2+ signal following a mock predator ambush (Figure 3J). We interpret these data together to indicate that locomotion does not correlate with an increase in calcium activity in SuMVGLUT2+::POA neurons, but that coping to a stressor does. This conclusion is further examined in supplemental Figure 5, including examining cross-correlation to test for temporally offset relationship between velocity and Ca2+ signal in SUMVGLUT2+::POA neurons.

The use of photostimulation only is unfortunate, it would have been really nice to see some inactivation of these neurons as well. This is because of the well-documented issues with being able to determine whether photostimulation is occurring in a physiological manner, and therefore makes certain data difficult to interpret. For instance, with regards to the 'active coping' behaviours - is this really the correct characterisation of what's going on? I wonder if the mice simply had developed immobile responding as a coping strategy but when they experience stimulation of these neurons that they find aversive, immobility is not sufficient to deal with the summative effects of the aversion from the swimming task as well as from the neuronal activation? An inactivation study would be more convincing.

We agree with the point of the reviewer, experiments demonstrating necessity of SUMVGLUT2+::POA neurons would have added to the story here. We carried out multiple experiments aimed at addressing questions about necessity of SuMVGLUT2+::POA neurons in stress coping behaviors, specifically the forced swim assay. Efforts included employing chemogenetic, optogenetic, and tetanus toxin-based methods. We observed no effects on locomotor activity or stress coping. These experiments are both technically difficult and challenging to interpret. Interpretation of negative results, as we obtained, is particularly difficult because of potential technical confounds. Selective targeting of SuMVGLUT2+::POA neurons for inhibition requires a process requiring three viral injections and two recombination steps, increasing variability and reducing the number of neurons impacted. Alternatively, photoinhibition targeting SuMVGLUT2+::POA cells can be done using Retro-AAV injected into POA and a fiber implant over SuM. We tried both approaches. Data obtained were difficult to interpret because of questions about adequate coverage of SuMVGLUT2+::POA population by virally expressed constructs and/or light spread arose. The challenge of adequate coverage to effectively prevent output from the targeted population is further confounded by challenges inherent in neural inhibition, specifically determining if the inhibition created at the cellular level is adequate to block output in the context of excitatory inputs or if neurons must be first engaged in a particular manner for inhibition to be effective. Baseline neural activity, release probability, and post-synaptic effects could all be relevant, which photo-inhibition will potentially not resolve. So, while the trend is to always show “necessary and sufficient” effects, we’ve tried nearly everything, and we simply cannot conclude much from our mixed results. There are also wellestablished problems with existing photo-inhibition methods, which while people use them and tout them, are often ignored. We have a lot of expertise in photo-inhibition optogenetics, and indeed have used it with some success, developed new methods, yet in this particular case we are unable to draw conclusions related to inhibition. People have experienced similar challenges in locus coeruleus neurons, which have very low basal activity, and inhibition with chemogenetics is very hard, as well as with optogenetic pump-based approaches, because the neurons fire robust rebound APs. We have spent almost 2.5 years trying to get this to work in this circuit because reviews have been insistent on this result for the paper to be conclusive. Unfortunately, it simply isn’t possible in our view until we know more about the cell types involved. This is all in spite of experience using the approach in many other publications.

We also employed less selective approaches, such as injecting AAV-DIO-tetanus toxin light chain (Tettox) constructs directly into SuM VGLUT2-Cre mice but found off target effects impacting animal wellbeing and impeding behavioral testing due viral spread to surrounding areas.

While we are disappointed for being unable to directly address questions about necessity of SuMVGLUT2+::POA neurons in active coping with experimental data, we were unable to obtain results allowing for clear interpretation across numerous other domains the reviewers requested. We also feel strongly that until we have a clear picture of the molecular cell type architecture in the SuM, and Cre-drivers to target subsets of neurons, this question will be difficult to resolve for any group. We are working now on RNAseq and related spatial transcriptomics efforts in the SuM and examining additional behavioral paradigm to resolve these issues, so stay tuned for future publications.

Accordingly, we avoid making statements relating to necessity in the manuscript. In spite of having several lines of physiological data with strong robust correlations behavior related to the SuMVGLUT2+::POA circuit.

Nose poke is only nominally instrumental as it cannot be shown to have a unique relationship with the outcome that is independent of the stimuli-outcome relationships (in the same way that a lever press can, for example). Moreover, there is nothing here to show that the behaviours are goal-directed.

Thank you for highlighting this point. Regarding goal-direct terminology, we removed this terminology from the manuscript. Since the mice perform highly selective (active vs inactive) port activation robustly across multiple days of training the behavior likely transitions to habitual behavior. We only tested the valuation of stimuli termination of the final day of training with time limited progressive ratio test. With respect to lever press versus active port activation, we are unclear how using a lever in this context would offer a different interpretation. Lever pressing may be more sensitive to changes in valuation when compared to nose poke port activation (Atalayer and Rowland 2008); however, in this study the focus of the operant behavior is separating innate behaviors for learned action–outcome instrumental learned behaviors for threat response (LeDoux and Daw 2018). The robust highly selective activation of the active port illustrated in Figure 6 fits as an action–outcome instrumental behavior wherein mice learn to engage the active but not inactive port to terminate photostimulation. The first activation of the port occurs through exploration of the arena but as demonstrated by the number of active port activations and the decline in time of the first active port engagement, mice expressing ChR2eYFP learn to engage the port to terminate the stimulation. To aid in illustrating this point we have added Supplemental Figure 7 showing active and inactive port activations for both Cre+ and Cre- mice. This adds clarity to high rate of selective port activation driven my stimulation of SUMVGLUT2+::POA neurons compared to controls. The elimination of goal directed and providing additional data narrows and supports one of the key points of the operant experiment.

With regards to Figure 1: This is a nice figure, but I wonder if some quantification of the pathways and their density might be helpful, perhaps by measuring the intensity of fluorescence in image J (as these are processes, not cell bodies that can be counted)? Mind you, they all look pretty dense so perhaps this is not necessary! However, because the authors are looking at projections in so-called 'stress-engaged regions', the amygdala seems conspicuous by its absence. Did the authors look in the amygdala and find no projections? If so it seems that this would be worth noting.

This is an interesting question but has proven to be a very technically challenging question. We consulted with several leaders who routinely use complimentary viral tracing methods in the field. We were unable to devise a method to provide a satisfactorily meaningful quantitative (as opposed to qualitative) approach to compare SUMVGLUT2+::POA to SuMVGLUT2+ projections. A few limitations are present that hinder a meaningful quantitative approach. One limitation was the need for different viral strategies to label the two populations. Labeling SuMVGLUT2+::POA neurons requires using VGLUT2-Flp mice with two injections into the POA and one into SuM. Two recombinase steps were required, reducing efficiency of overlap. This combination of viral injections, particularly the injections of RetroAAVs in the POA, can induce significant quantitative variability due to tropism, efficacy, and variability of retro-viral methods, and viral infection generally. These issues are often totally ignored in similar studies across the “neural circuit” landscape, but it doesn’t make them less relevant here.

Although people do this in the field, and show quantification, we actually believe that it can be a quite misleading read-out of functionally relevant circuitry, given that neurotransmitter release ultimately is amplified by receptors post-synaptically, and many examples of robust behavioral effects have been observed with low fiber tracing complimentary methods (McCall, Siuda et al. 2017). In contrast, the broader SuMVGLUT2+ population was labeled using a single injection into the SuM. This means there like more efficient expression of the fluorophore. Additionally, in areas that contain terminals and passing fibers understanding and interpreting fluorescent signal is challenging. Together, these factors limit a meaningful quantitative comparison and make an interpretation difficult to make. In this context, we focused on a conservative qualitative presentation to demonstrate two central points. That (1) SuMVGLUT2+::POA neurons are subset of SuMVGLUT2+ neurons that project to specific areas and that exclude dentate gyrus, and they (2) arborize extensively to multiple areas which have be linked to threat responses. We agree that there is much to be learned about how different populations in SuM connect to targets in different regions of the brain and to continue to examine this question with different techniques. A meaningful quantitative study comparing projections is technically complex and, we feel, beyond our ability for this study.

Also, for the reasons above we do not believe that quantification provides exceptional clarity with respect to the putative function of the circuit, glutamate released, or other cotransmitters given known amplification at the post-synaptic side of the circuit.

With regard to the amygdala, other studies on SuM projections have found efferent projections to amygdala (Ottersen, 1980; Vertes, 1992). In our study we were unable to definitively determine projections from SuMVGLUT2+::POA neurons to amygdala, which if present are not particularly dense. For this reason we were conservative and do not comment on this particular structure.

I would suggest removing the term goal-directed from the manuscript and just focusing on the active vs. passive distinction.

We removed the use of goal-directed. Thank you for helping us clarify our terminology.

The effect observed in Figure 7I is interesting, and I'm wondering if a rebound effect is the most likely explanation for this. Did the authors inhibit the VGAT neurons in this region at any other times and observe a similar rebound? If such a rebound was not observed it would suggest that it is something specific about this task that is producing the behaviour. I would like it if the authors could comment on this.

We agree that results showing the change in coping strategy (passive to active) in forced swim after but not during stimulation of SuMVGAT+ neurons is quite interesting (Figure 7I). This experiment activated SuMVGAT+ neurons during a section of the forced swim assay and mice showed a robust shift to mobility after the stimulation of SuMVGAT+ neurons stopped. We did not carry out inhibition of SuMVGAT+ neurons in this manuscript. As the reviewer suggested, strong inhibition of local SuM neurons, including SUMVGLUT2+::POA neurons, could lead to rebound activity that may shift coping behaviors in confusing ways. We agree this is an interesting idea but do not have data to support the hypothesis further at this time.

**Reviewer 2**
(1) These are very difficult, small brain regions to hit, and it is commendable to take on the circuit under investigation here. However, there is no evidence throughout the manuscript that the authors are reliably hitting the targets and the spread is comparable across experiments, groups, etc., decreasing the significance of the current findings. There are no hit/virus spread maps presented for any data, and the representative images are cropped to avoid showing the brain regions lateral and dorsal to the target regions. In images where you can see the adjacent regions, there appears expression of cell bodies (such as Supp 6B), suggesting a lack of SuM specificity to the injections.

We agree with the reviewer that the areas studied are small and technically challenging to hit. This was one of driving motivations for using multiple tools in tandem to restrict the area targeted for stimulation. Approaches included using a retrograde AAVs to express ChR2eFYP in SUMVGLUT2+::POA neurons; thereby, restricting expression to VGLUT2+ neurons that project to the POA. Targeting was further limited by placement of the optic fiber over cell bodies on SuM. Thus, only neurons that are VGLUT2+, project to the POA, and were close enough to the fiber were active by photostimulation. Regrettably, we were not able to compile images from mice where the fiber was misplaced leading to loss of behavioral effects. We would have liked to provide that here to address this comment. Unfortunately, generating heat maps for injections is not possible for anatomic studies that use unlabeled recombinase as part of an intersectional approach. Also determining the point of injection of a retroAAV can be difficult to accurately determine its location because neurons remote to injection site and their processes are labeled.

Experiments described in Supplemental Figure 6B on VGAT neurons in SuM were designed and interpreted to support the point that SUMVGLUT2+::POA neurons are a distinct population that does not overlap with GABAergic neurons. For this point it is important that we targeted SuM, but highly confined targeting is not needed to support the central interpretation of the data. We do see labeling in SuM in VGAT-Cre mice but photo stimulation of SuMVGAT+ neurons does not generate the behavioral changes seen with activation of SUMVGLUT2+::POA neurons. As the reviewer points out, SuM is small target and viral injection is likely to spread beyond the anatomic boundaries to other VGAT+ neurons in the region, which are not the focus here. The activation would be restricted by the spread of light from the fiber over SuM (estimated to be about a 200um sphere in all directions). We did not further examine projections or localization of VGAT+ neurons in this study but focused on the differential behavioral effects of SUMVGLUT2+::POA neurons.

(2) In addition, the whole brain tracing is very valuable, but there is very little quantification of the tracing. As the tracing is the first several figures and supp figure and the basis for the interpretation of the behavior results, it is important to understand things including how robust the POA projection is compared to the collateral regions, etc. Just a rep image for each of the first two figures is insufficient, especially given the above issue raised. The combination of validation of the restricted expression of viruses, rep images, and quantified tracing would add rigor that made the behavioral effects have more significance.For example, in Fig 2, how can one be sure that the nature of the difference between the nonspecific anterograde glutamate neuron tracing and the Sum-POA glutamate neuron tracing is real when there is no quantification or validation of the hits and expression, nor any quantification showing the effects replicate across mice? It could be due to many factors, such as the spread up the tract of the injection in the nonspecific experiment resulting in the labeling of additional regions, etc.Relatedly, in Supp 4, why isn’t C normalized to DAPI, which they show, or area? Similar for G what is the mcherry coverage/expression, and why isn’t Fos normalized to that?

Thank you for highlighting the importance of anatomy and the value of anatomy. Two points based on the anatomic studies are central to our interpretation of the experimental data. First, SUMVGLUT2+::POA are a distinct population within the SuM. We show this by demonstrating they are not GABAergic and that they do not project to dentate gyrus. Projections from SuM to dentate gyrus have been described in multiple studies (Boulland et al., 2009; Haglund et al., 1987; Hashimotodani et al., 2018; Vertes, 1992) and we demonstrate them here for SuMVGLUT2+ cells. Using an intersectional approach in VGLUT2-Flp mice we show SUMVGLUT2+::POA neurons do not project to dentate gyrus. We show cell bodies of SUMVGLUT2+::POA neurons located in SuM across multiple figures including clear brain images. Thus, SUMVGLUT2+::POA neurons are SuM neurons that do not project to dentate gyrus, are not GABAergic, send projections to a distinct subset of targets, most notably excluding dentate gyrus. Second, SUMVGLUT2+::POA neurons arborize sending projections to multiple regions. We show this using a combinatorial genetic and viral approach to restrict expression of eYFP to only neurons that are in SuM (based on viral injection), project to the POA (based on retrograde AAV injection in POA), and VGLUT2+ (VGLUT2-Flp mice). Thus, any eYFP labeled projection comes from SUMVGLUT2+::POA neurons. We further confirmed projections using retroAAV injection into areas identified using anterograde approaches (Supplemental Figure 2). As discussed above in replies to Reviewer 1, we feel limitations are present that preclude meaningful quantitative analysis. We thus opted for a conservative interpretation as outlined.

Prior studies have shown efferent projections from SuM to many areas, and projections to dentate gyrus have received substantial attention (Bouland et al., 2009; Haglund, Swanson, and Kohler, 1984; Hashimotodani et al., 2018; Soussi et al., 2010; Vertes, 1992; Pan and McNaugton, 2004). We saw many of the same projections from SuMVGLUT2+ neurons. We found no projections from SUMVGLUT2+::POA neurons to dentate gyrus (Figure 2). Our description of SuM projection to dentate gyrus is not new but finding a population of neurons in SuM that does not project to dentate gyrus but does project to other regions in hippocampus is new. This finding cannot be explained by spread of the virus in the tract or non-selective labeling.

(3) The authors state that they use male and female mice, but they do not describe the n’s for each experiment or address sex as a biological variable in the design here. As there are baseline sex differences in locomotion, stress responses, etc., these could easily factor into behavioral effects observed here.

Sex specific effects are possible; however, the studies presented here were not designed or powered to directly examine them. A point about experimental design that helps mitigate against strong sex dependent effect is that often the paradigm we used examined baseline (pre-stimulation) behavior, how behavior changed during stimulation, and how behavior returned (or not) to baseline after stimulation. Thus, we test changes in individual behaviors. Although we had limited statistical power, we conducted analyses to examine the effects of sex as variable in the experiments and found no differences among males and females.

(4) In a similar vein as the above, the authors appear to use mice of different genotypes (however the exact genotypes and breeding strategy are not described) for their circuit manipulation studies without first validating that baseline behavioral expression, habituation, stress responses are not different. Therefore, it is unclear how to interpret the behavioral effects of circuit manipulation. For example in 7H, what would the VGLUT2-Cre mouse with control virus look like over time? Time is a confound for these behaviors, as mice often habituate to the task, and this varies from genotype to genotype. In Fig 8H, it looks like there may be some baseline differences between genotypes- what is normal food consumption like in these mice compared to each other? Do Cre+ mice just locomote and/or eat less? This issue exists across the figures and is related to issues of statistics, potential genotype differences, and other experimental design issues as described, as well as the question about the possibility of a general locomotor difference (vs only stress-induced). In addition, the authors use a control virus for the control groups in VGAT-Cre manipulation studies but do not explain the reasoning for the difference in approach.

Thank you for highlighting the need for greater clarity about the breeding strategies used and for these related questions. We address the breeding strategy and then move to address the additional concerns raised. We have added details to the methods section to address this point. For VGLUT2-Cre mice we use litter mates controls from Cre/WT x WT/WT cross. The VGLUT2-Cre line (RRID:IMSR_JAX:028863) (Vong L , et al. 2011) used here been used in many other reports. We are not aware of any reports indicating a phenotype associated with the addition of the IRES-Cre to the Slc17a6 loci and there is no expected impact of expression of VGLUT2. Also, we see in many of the experiments here that the baseline (Figures 4, 5, and 7) behaviors are not different between the Cre+ and Cre- mice. For VGAT-Cre mice we used a different breeding strategy that allowed us to achieve greater control of the composition of litters and more efficient cohorts cohort. A Cre/Cre x WT/WT cross yielded all Cre/WT litters. The AAV injected, ChR2eYFP or eYFP, allowed us to balance the cohort.

Regarding Figure 7H, which shows time immobile on the second day of a swim test, data from the Cre- mice demonstrate the natural course of progression during the second day of the test. The control mice in the VGAT-Cre cohort (Figure 7I) have similar trend. The change in behavior during the stimulation period in the Cre+ mice is caused by the activation of SUMVGLUT2+::POA neurons. The behavioral shift largely, but not completely, returns to baseline when the photostimulation stops. We have no reason to believe a VGLUT2-Cre+ mouse injected with control AAV to express eYFP would be different from WT littermate injected with AVV expressing ChR2eYFP in a Cre dependent manner.

Turning to concerns related to 8H, which shows data from fasted mice quantify time spent interacting with food pellet immediately after presentation of a chow pellet, we found no significant difference between the control and Cre+ mice. We unaware of any evidence indicating that the two groups should have a different baseline since the Cre insertion is not expected to alter gene expression and we are unaware of reports of a phenotype relating to feeding and the presence of the transgene in this mouse line. Even if there were a small baseline shift this would not explain the large abrupt shift induced by the photostimulation. As noted above, we saw shifts in behavior abruptly induced by the initiation of photostimulation when compared to baseline in multiple experiments. This shift would not be explained by a hypothetical difference in the baseline behaviors of litter mates.

(5) The statistics used throughout are inappropriate. The authors use serial Mann-Whitney U tests without a description of data distributions within and across groups. Further, they do not use any overall F tests even though most of the data are presented with more than two bars on the same graph. Stats should be employed according to how the data are presented together on a graph. For example, stats for pre-stim, stim, and post-stim behavior X between Cre+ and Cre- groups should employ something like a two-way repeated measures ANOVA, with post-hoc comparisons following up on those effects and interactions. There are many instances in which one group changes over time or there could be overall main effects of genotype. Not only is serially using Mann-Whitney tests within the same panel misleading and statistically inaccurate, but it cherry-picks the comparisons to be made to avoid more complex results. It is difficult to comprehend the effects of the manipulations presented without more careful consideration of the appropriate options for statistical analysis.

We thank the reviewer for pointing this out and suggesting alterative analyses, we agree with the assessment on this topic. Therefore, we have extensively revised the statical approach to our data using the suggested approach. Reviewer 1 also made a similar comment, and we would like to point to our reply to reviewer 1’s second point in regard to what we changed and added to the new statistical analyses. Further, we have added a full table detailing the statical values for each figure to the paper.

Conceptual:(6) What does the signal look like at the terminals in the POA? Any suggestion from the data that the projection to the POA is important?

This is an interesting question that we will pursue in future investigations into the roles of the POA. We used the projection to the POA from SuM to identify a subpopulation in SuM and we were surprised to find the extensive arborization of these neurons to many areas associated with threat responses. We focused on the cell bodies as “hubs” with many “spokes”. Extensive studies are needed to understand the roles of individual projections and their targets. There is also the hypothetical technical challenge of manipulating one projection without activating retrograde propagation of action potentials to the soma. At the current time we have no specific insights into the roles of the isolated projection to POA. Interpretation of experiments activating only “spoke” of the hub would be challenging. Simple terminal stimulation experiments are challenged by the need to separate POA projections from activation of passing fibers targeting more anterior structures of the accumbens and septum.

(7) Is this distinguishing active coping behavior without a locomotor phenotype? For example, Fig. 5I and other figure panels show a distance effect of stimulation (but see issues raised about the genotype of comparison groups). In addition, locomotor behavior is not included for many behaviors, so it is hard to completely buy the interpretation presented.

We agree with the reviewer and thank them for highlighting this fundamental challenge in studies examining active coping behaviors in rodents, which requires movement. Additionally, actively responding to threatening stressors would include increased locomotor activity. Separation of movement alone from active coping can be challenging. Because of these concerns we undertook experiments using diverse behavioral paradigms to examine the elicited behaviors and the recruitment of SuMVGLUT2+::POA neurons to stressors. We conducted experiments to directly examine behaviors evoked by photoactivation of SuMVGLUT2+::POA. In these experiments we observed a diversity of behaviors including increased locomotion and jumping but also treading/digging (Figure 4). These are behaviors elicited in mice by threatening and noxious stimuli. An Increase of running or only jumping could signify a specific locomotor effect, but this is not what was observed. Based on these behaviors, we expected to find evidence of increase movement in open field (Figure 5G-I) and light dark choice (Figure 5J-L) assays. For many of the assays, reporting distance traveled is not practical. An important set of experiments that argues against a generic increase in locomotion is the operant behavior experiments, which require the animal to engage in a learned behavior while receiving photostimulation of SuMVGLUT2+::POA neurons (Figure 6). This is particularly true for testing using a progressive ratio when the time of ongoing photostimulation is longer, yet animals actively and selectively engage the active port (Figure 6G-H). Further, we saw a shift in behavioral strategy induce by photoactivation in forced swim test (Figure 7H). Thus, activation of SUMVGLUT2+::POA neurons elicited a range of behaviors that included swimming, jumping, treading, and learned response, not just increased movement. Together these data strongly argue that SuMVGLUT2+::POA neurons do not only promote increased locomotor behavior. We interpret these data together with the data from fiber photometry studies to show SuMVGLUT2+::POA neurons are recruited during acute stressors, contribute to aversive affective component of stress, and promote active behaviors without constraining the behavioral pattern.

Regarding genotype, we address this in comments above as well but believe that clarifying the use of litter mates, the extensive use of the VGLUT2-Cre line by multiple groups, and experimental design allowing for comparison to baseline, stimulation evoked, and post stimulation behaviors within and across genotypes mitigate possible concerns relating to the genotype.

(8) What is the role of GABA neurons in the SuM and how does this relate to their function and interaction with glutamate neurons? In Supp 8, GABA neuron activation also modulates locomotion and in Fig 7 there is an effect on immobility, so this seems pretty important for the overall interpretation and should probably be mentioned in the abstract.

Thank you for noting these interesting findings. We added text to highlight these findings to the abstract. Possible roles of GABAergic neurons in SuM extend beyond the scope of the current study particularly since SuM neurons have been shown to release both GABA and glutamate (Li Y, Bao H, Luo Y, et al. 2020, Root DH, Zhang S, Barker DJ et al. 2018). GABAergic neurons regulate dentate gyrus (Ajibola MI, Wu JW, Abdulmajeed WI, Lien CC 2021), REM sleep (Billwiller F, Renouard L, Clement O, Fort P, Luppi PH 2017), and novelty processing Chen S, He L, Huang AJY, Boehringer R et al. 2020. The population of exclusively GABAergic vs dual neurotransmitter neurons in SuM requires further dissection to be understood. How they may relate to SUMVGLUT2+::POA neurons require further investigation.

Questions about figure presentation:(9) In Fig 3, why are heat maps shown as a single animal for the first couple and a group average for the others?

Thank you for highlighting this point for further clarification. We modified the labels in the figure to help make clear which figures are from one animal across multiple trials and those that are from multiple animals. In the ambush assay each animal one had one trial, to avoid habituation to the mock predator. Accordingly, we do not have multiple trials for each animal in this test. In contrast, the dunk assay (10 trial/animal) and the shock (5 trials/animal) had multiple trials for each animal. We present data from a representative animal when there are multiple trials per animal and the aggerate data.

Why is the temporal resolution for J and K different even though the time scale shown is the same?

Thank you for noticing this error carried forward from a prior draft of the figure so we could correct it. We replaced the image in 3J with a more correctly scaled heatmap.

What is the evidence that these signal changes are not due to movement per se?

Thank you for the question. There are two points of evidence. First, all the 465 nm excitation (Ca2+ dependent) data was collected in interleaved fashion with 415 nm (isosbestic) excitation data. The isosbestic signal is derived from GCaMP emission but is independent of Ca2+ binding (Martianova E, Aronson S, Proulx CD. 2019). This approach, time-division multiplexing, can correct calcium-dependent for changes in signal most often due to mechanical change. The second piece of evidence is experimental. Using multiple cohorts of mice, we examined if the change in Ca2+ signal was correlated with movement. We used the threshold of velocity of movement seen following the ambush. We found no correlation between high velocity movements and Ca2+ signal (Figure 3K) including cross correlational analysis (Supplemental figure 5). Based on these points together we conclude the change in the Ca2+ signal in SUMVGLUT2+::POA neurons is not due to movement induced mechanical changes and we find no correlation to movement unless a stressor is present, i.e. mock predator ambush or forced swim. Further, the stressors evoke very different locomotor responses fleeing, jumping, or swimming.

(10) In Fig 4, the authors carefully code various behaviors in mice. While they pick a few and show them as bars, they do not show the distribution of behaviors in Cre- vs Cre+ mice before manipulation (to show they have similar behaviors) or how these behaviors shift categories in each group with stimulation. Which behaviors in each group are shifting to others across the stim and post-stim periods compared to pre-stim?

This is an important point. We selected behaviors to highlight in Figure4 C-E because these behaviors are exhibited in response to stress (De Boer & Koolhaas, 2003; van Erp et al., 1994). For the highlighted behaviors, jumping, treading/digging, grooming, we show baseline (pre photostimulation), stimulation, and post stimulation for Cre+ and Cre- mice with the values for each animal plotted. We show all nine behaviors as a heat map in Figure 4B. The panels show changes that may occur as a function of time and show changes induced by photostimulation.

The heatmaps demonstrate that photostimulation of SUMVGLUT2+::POA neurons causes a suppression of walking, grooming, and immobile behaviors with an increase in jumping, digging/treading, and rapid locomotion. After stimulation stops, there is an increase in grooming and time immobile. The control mice show a range of behaviors with no shifts noted with the onset or termination of photostimulation.

Of note, issues of statistics, genotype, and SABV are important here. For example, the hint that treading/digging may have a slightly different pre-stim basal expression, it seems important to first evaluate strain and sex differences before interpreting these data.

We examined the effects of sex as a biological variable in the experiments reported in the manuscript and found no differences among males and females in any of the experiments where we had enough animals in each sex (minimum of 5 mice) for meaningful comparisons. We did this by comparing means and SEM of males and females within each group (e.g. Cre+ males vs Cre+ female, Cre- males vs Cre- females) and then conducted a t-test to see if there was a difference. For figures that show time as a variable (e.g Figure 6C-E), we compared males and females with time x sex as main factors and compared them (including multiple comparisons if needed). We found no significant main effects or interactions between males and females. Because of this, and to maximize statistical power, we decided to move forward to keep males and females together in all the analyses presented in the manuscript. It is worth noting also that the core of the experimental design employed is a change in behavior caused by photostimulation. The mice are also the same strain with only difference being the modification to add an IRES and sequence for Cre behind the coding sequence of the Slc17A6 (VGLUT2) gene.

(11) Why do the authors use 10 Hz stimulation primarily? is this a physiologically relevant stim frequency? They show that they get effects with 1 Hz, which can be quite different in terms of plasticity compared to 10 Hz.

Thank you for the raising this important question. Because tests like open field and forced swim are subject to habituation and cannot be run multiple times per animal a test frequency was needed to use across multiple experiments for consistency. The frequency of 10Hz was selected because it falls within the rate of reported firing rates for SuM neurons (Farrel et al., 2021; Pedersen et al., 2017) and based on the robust but sub maximal effects seen in the real-time place preference assays. Identification of the native firing rates during stress response would be ideal but gathering this data for the identified population remains a dauting task.

(12) In Fig 5A-F, it is unclear whether locomotion differences are playing a role. Entrances (which are low for both groups) are shown but distance traveled or velocity are not.In B, there is no color in the lower left panel. where are these mice spending their time? How is the entirety of the upper left panel brighter than the lower left? If the heat map is based on time distribution during the session, there should be more color in between blue and red in the lower left when you start to lose the red hot spots in the upper left, for example. That is, the mice have to be somewhere in apparatus. If the heat map is based on distance, it would seem the Cre- mice move less during the stim.

We appreciate the opportunity to address this question, and the attention to detail the reviewer applied to our paper. In the real time place preference test (RTPP) stimulation would only be provided while the animal was on the stimulation side. Mice quickly leave the stimulation side of the arena, as seen in the supplemental video, particularly at the higher frequencies. Thus, the time stimulation is applied is quite low. The mice often retreat to a corner from entering the stimulation side during trials using higher frequency stimulation. Changing locomotor activity along could drive changes in the number entrances but we did not find this. In regard to the heat map, the color scale is dynamically set for each of the paired examples that are pulled from a single trial. To maximize the visibility between the paired examples the color scale does not transfer between the trials. As a result, in the example for 10 Hz the mouse spent a larger amount of time in the in the area corresponding to the lower right corner of the image and the maximum value of the color scale is assigned to that region. As seen in the supplemental video, mice often retreated to the corner of the non-stimulation side after entering the stimulation side. The control animal did not spend a concentrated amount of time in any one region, thus there is a lack of warmer colors. In contrast the baseline condition both Cre+ and Cre- mice spent time in areas disturbed on both sides of arena, as expected. As a result, the maximum value in the heat map is lower and more area are coded in warmer colors allowing for easier visual comparison between the pair. Using the scale for the 10 Hz pair across all leads to mostly dark images. We considered ways to optimized visualization across and within pairs and focused on the within pair comparison for visualization.

(13) By starting with 1 hz, are the experimenters inducing LTD in the circuit? what would happen if you stop stimming after the first epoch? Would the behavioral effect continue? What does the heat map for the 1 hz stim look like?Relatedly, it is a lot of consistent stimulation over time and you likely would get glutamate depletion without a break in the stim for that long.

Thank you for the opportunity to add clarity around this point regarding the trials in RTPP testing. Importantly, the trials were not carried out in order of increasing frequency of stimulation, as plotted. Rather, the order of trials was, to the extent possible with the number of mice, counterbalanced across the five conditions. Thus, possible contribution of effects of one trial on the next were minimized by altering the order of the trials.

We have added a heat map for the 1 Hz condition to figure 5B.

For experiments on RTPP the average stimulation time at 10Hz was less than 10 seconds per event. As a result, the data are unlikely to be affected by possible depletion of synaptic glutamate. For experiments using sustained stimulation (open field or light dark choice assays) we have no clear data to address if this might be a factor where 10Hz stimulation was applied for the entire trial.

(14) In Fig 6, the authors show that the Cre- mice just don't do the task, so it is unclear what the utility of the rest of the figure is (such as the PR part). Relatedly, the pause is dependent on the activation, so isn't C just the same as D? In G and H, why ids a subset of Cre+ mice shown?Why not all mice, including Cre- mice?

Thank you for the opportunity to improve the clarity of this section. A central aspect of the experiments in Figure 6 is the aversiveness of SUMVGLUT2+::POA neuron photostimulation, as shown in Figure 5B-F. The aversion to photostimulation drives task performance in the negative reinforcer paradigm. The mice perform a task (active port activation) to terminate the negative reinforcer (photostimulation of SuMVGLUT2+::POA neurons). Accordingly, control mice are not expected to perform the task because SuMVGLUT2+::POA neurons are not activated and, thus the mice are not motivated to perform the task.

A central point we aim to covey in this figure is that while SuMVGLUT2+::POA neurons are being stimulated, mice perform the operant task. They selectively activated the active port (Supplemental Figure 7). As expected, control mice activate the active port at a low level in the process of exploring the arena. This diminishes on subsequent trials as mice habituate to the arena (Figure 6D). The data in Figures 6 C and D are related but can be divergent. Each pause in stimulation requires a port activation of a FR1 test but the number of port activations can exceed the pauses, which are 10 seconds long, if the animal continues to activate the port. Comparing data in Figures 6 C and D revels that mice generally activated the port two to three times for each pause earned with a trend towards greater efficiency on day 4 with more rewards and fewer activations.

The purpose of the progressive ratio test is to examine if photostimulation of SuMVGLUT2+::POA continues to drive behavior as the effort required to terminate the negative stimuli increases. As seen in Figures 6 G and H, the stimulation of SuMVGLUT2+::POA neurons remains highly motivating. In the 20-minute trial we did not find a break point even as the number of port activations required to pause the stimulation exceed 50. We do not show the Cre- mice is Figure 6G and H because they did not perform the task, as seen in Figure 6F. For technical reasons in early trials, we have fully timely time stamped data for rewards and port activations from a subset of the Cre+ mice. Of note, this contains both the highest and lowest performing mice from the entire data set.

Taken together, we interpret the results of the operant behavioral testing as demonstrating that SuMVGLUT2+::POA neuron activation is aversive, can drive performance of an operant tasks (as opposed to fixed escape behaviors), and is highly motivating.

(15) In Fig 7, what does the GCaMP signal look like if aligned to the onset of immobility? It looks like since the hindpaw swimming is short and seems to precede immobility, and the increase in the signal is ramping up at the onset of hindpaw swimming, it may be that the calcium signal is aligned with the onset of immobility.What does it look like for swimming onset?In I, what is the temporal resolution for the decrease in immobility? Does it start prior to the termination of the stim, or does it require some elapsed time after the termination, etc?

Thank for the opportunity to addresses these points and improve that clarity of our interpretation of the data. Regarding aligning the Ca2+ signal from fiber photometry recordings to swimming onset and offset, it is important to note that the swimming bouts are not the same length. As a result, in the time prior to alignment to offset of behaviors animals will have been swimming for different lengths of time. In Figure 7 C, we use the behavioral heat map to convey the behavioral average. Below we show the Ca2+ dependent signal aligned at the offset of hindpaw swim for an individual mouse (A) and for the total cohort (B). This alignment shows that the Ca2+ dependent signal declines corresponding to the termination of hindpaw swimming.Because these bouts last less than the total the widow shown, the data is largely included in Figure 7 C and D, which is aligned to onset. Due to the nuance of the difference is the alignment and the partial redundancy, we elected to include the requested alignment to swimming offset in the reply rather in primary figure.

**Author response image 1. sa2fig1:** 

Turning to the question regarding swimming onset, the animals started swimming immediately when placed in the water and maintained swimming and climbing behaviors until shifting behaviors as illustrated in Figure 7A and B. During this time the Ca2+-dependent signal was elevated but there is only one trial per animal. This question can perhaps be better addressed in the dunk assay presented in Figure 3C, F and G and Supplemental Figure 4 H and I. Here swimming started with each dunk and the Ca2+ signal increased.

Regarding the question for about figure 7I. We scored for entire periods (2 mins) in aggerate. We noted in videos of the behavior test that there was an abrupt decrease in immobility tightly corresponding to the end of stimulation. In a few animals this shift occurred approximately 15-20s before the end of stimulation. This may relate to the depletion of neurotransmitter as suggested by the reviewer.

**Reviewer 3**
Major points(1) Results in Figure 1 suggested that SuM-Vglu2::POA projected not only POA but also to the diverse brain regions. We can think of two models which account for this. One is that homogeneous populations of neurons in SuM-Vglu2::POA have collaterals and innervated all the efferent targets shown in Figure 1. Another is to think of distinct subpopulations of neurons projecting subsets of efferent targets shown in Figure 1 as well as POA. It is suggested to address this by combining approaches taken in experiments for Figure 1 and Supplemental Figure 2.

Thank you for raising this interesting point. We have attempted combining retroAAV injections into multiple areas that receive projections from SUMVGLUT2+::POA neurons. However, we have found the results unsatisfactory for separating the two models proposed. Using eYFP and tdTomato expressing we saw some overlapping expressing in SuM. We are not able to conclude if this indicates separate populations or partial labeling of a homogenous populations. A third option seems possible as well. There could be a mix of neurons projecting to different combinations of downstream targets. This seems particularly difficult to address using fluorophores. We are preparing to apply additional methodologies to this question, but it extends beyond the scope of this manuscript.

(2) Since the authors drew a hypothetical model in which the diverse brain regions mediate the effect of SuM-Vglu2::POA activation in behavioral alterations at least in part, examination of the concurrent activation of those brain regions upon photoactivation of SuM-Vglu2::POA. This must help the readers to understand which neural circuits act upon the induction of active coping behavior under stress.

Thank you for raising this important point. We agree that activating glutamatergic neurons should lead to activation of post synaptic neurons in the target regions. Delineating this in vivo is less straight forward. Doing so requires much greater knowledge of post synaptic partners of SUMVGLUT2+::POA neurons. There are a number of issues that would need to be accounted for. Undertaking two color photo stimulation plus fiber photometry is possible but not a technical triviality. Further, it is possible that we would measure Ca2+ signals in neurons that have no relevant input or that local circuits in a region may shape the signal. We would also lack temporal resolution to identify mono-postsynaptic vs polysynaptic connections. Thus, we would struggle to know if the change in signal was due to the excitatory input from SuM or from a second region. At present, we remain unclear on how to pursue this question experimentally in a manner that is likely to generate clearly interpretable results.

(3) In Figure 4, "active coping behaviors" must be called "behaviors relevant to the active behaviors" or "active coping-like behaviors", since those behaviors were in the absence of stressors to cope with.

Thank you for the suggestion on how to clarify our terminology. We have adopted the active coping-like term.

(4) For the Dunk test, it is suggested to describe the results and methods more in detail, since the readers would be new to it. In particular, the mice could change their behavior between dunks under this test, although they still showed immobility across trials as in Supplemental Figure 4I. Since neural activity during the test was summarized across trials as in Figure 3, it is critical to examine whether the behavior changes according to time.

Thank you for identifying this opportunity to improve our manuscript. We have expanded and added a detailed description of the dunk test in the methods section.

As for Supplemental Figure 4I, we apologize for the confusion because the purpose of this figure is to show that mice remained mobile for the entire 30-second dunk trial. This did not appreciably change over the 10 trials. We have revised this figure to plot both immobile and mobile time to achieve greater clarity on this point.

Minor pointsTyposIn Figure 1, please add a serotype of AAVs to make it compatible with other figures and their legends.In the main text and Figure 2K, the authors used MHb/LHb and mHb/lHb in a mixed fashion. Please make them unified.In the figure legend of Figure 6, change "SuMVGLUT2+::POA neurons drive" to "SuMVGLUT2+::POA neurons " in the title.In line 86, please change "Retro-AAV2-Nuc-flox(mCherry)-eGFP" to "AAV5-Nuc-flox(mCherry)eGFP".In line 80, please change "Positive controls" to "As positive controls, ".

Thank you for taking the time and making the effort to identify and call these out. We have corrected them.